# RESIDUAL-MPPI: ONLINE POLICY CUSTOMIZATION FOR CONTINUOUS CONTROL

Pengcheng Wang[1*], Chenran Li[1*], Catherine Weaver[1], Kenta Kawamoto[3], Masayoshi Tomizuka[1], Chen Tang[2†], and Wei Zhan[1]

[1]Department of Mechanical Engineering, University of California, Berkeley.
[2]Department of Computer Science, University of Texas at Austin.
[3]Sony Research Inc., Japan.

## ABSTRACT

Policies developed through Reinforcement Learning (RL) and Imitation Learning (IL) have shown great potential in continuous control tasks, but real-world applications often require adapting trained policies to unforeseen requirements. While fine-tuning can address such needs, it typically requires additional data and access to the original training metrics and parameters. In contrast, an online planning algorithm, if capable of meeting the additional requirements, can eliminate the necessity for extensive training phases and customize the policy without knowledge of the original training scheme or task. In this work, we propose a generic online planning algorithm for customizing continuous-control policies at the execution time, which we call *Residual-MPPI*. It can customize a given prior policy on new performance metrics in few-shot and even zero-shot online settings, given access to the prior action distribution alone. Through our experiments, we demonstrate that the proposed Residual-MPPI algorithm can accomplish the few-shot/zero-shot online policy customization task effectively, including customizing the champion-level racing agent, Gran Turismo Sophy (GT Sophy) 1.0, in the challenging car racing scenario, Gran Turismo Sport (GTS) environment. Code for MuJoCo experiments is included in the supplementary and will be open-sourced upon acceptance. Demo videos and code are available on our website: https://sites.google.com/view/residual-mppi.

## 1 INTRODUCTION

Policy learning algorithms such as Reinforcement Learning (RL) and Imitation Learning (IL) have been widely employed to synthesize parameterized control policies for a wide range of real-world motion planning and decision-making problems (Tang et al., 2024), such as navigation (Mirowski et al., 2016; Francis et al., 2020), manipulation (Andrychowicz et al., 2020; Zhang et al., 2018; Mandlekar et al., 2021; Rajeswaran et al., 2017) and locomotion (Peng & Van De Panne, 2017a;b; Gangapurwala et al., 2022). In practice, real-world applications often impose additional requirements on the trained policies beyond those established during training, which can include novel goals (Rhinehart et al., 2018), specific behavior preferences (Ziegler et al., 2019), and stringent safety criteria (Lu et al., 2023). Retraining a new policy network whenever a new additional objective is encountered is both expensive and inefficient, as it may demand extensive training efforts. To enable flexible deployment, it is thereby crucial to develop sample-efficient algorithms for synthesizing new policies that meet additional objectives while preserving the characteristics of the original policy (Lu et al., 2023; Harmel et al., 2023).

Recently, Li et al. (2024) introduced a new problem setting termed *policy customization*, which provides a principled approach to address the aforementioned challenge. In policy customization, the objective is to develop a new policy given a prior policy, ensuring that the new policy: 1) retains the

---
* Equal Contribution   † Corresponding Author

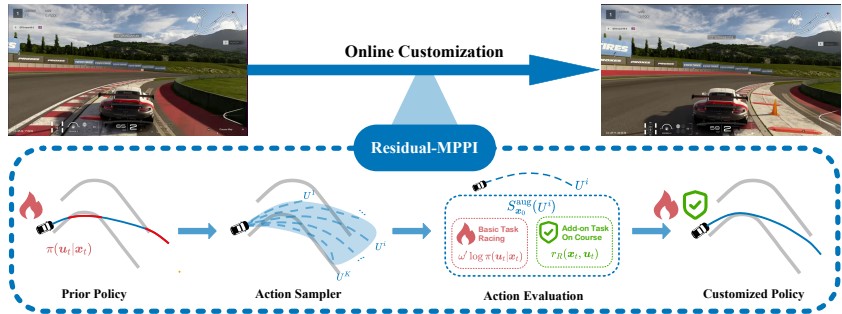

Figure 1: Overview of the proposed algorithm. In each planning loop, we utilize the prior policy to generate samples and then evaluate them through both the log likelihood of the prior policy and an add-on reward to obtain the customized actions. More details are in Sec. 3. In the experiments, we demonstrate that Residual-MPPI can accomplish the online policy customization task effectively, even in a challenging GTS environment with the champion-level racing agent, GT Sophy 1.0.

properties of the prior policy, and 2) fulfills additional requirements specified by a given downstream task. As an initial solution, the authors proposed the Residual Q-learning (RQL) framework. For discrete action spaces, RQL can leverage maximum-entropy Monte Carlo Tree Search (Xiao et al., 2019) to customize the policy *online*, meaning at the inference time without training. In contrast, for continuous action spaces, RQL offers a solution based on Soft Actor-Critic (SAC) (Haarnoja et al., 2018) to train a customized policy leveraging the prior policy before the real execution. While SAC-based RQL is more sample-efficient than training a new policy from scratch, the additional training steps that are required may still be expensive and time-consuming.

In this work, we propose *online policy customization* for *continuous action spaces* that eliminates the need for additional policy training. We leverage Model Predictive Path Integral (MPPI) (Williams et al., 2017), a sampling-based model predictive control (MPC) algorithm. Specifically, we propose *Residual-MPPI*, which integrates RQL into the MPPI framework, resulting in an online planning algorithm that customizes the prior policy at the execution time. The policy performance can be further enhanced by iteratively collecting data online to update the dynamics model. Our experiments in MuJoCo demonstrate that our method can effectively achieve zero-shot policy customization with a provided offline trained dynamics model. Furthermore, to investigate the scalability of our algorithm in complex environments, we evaluate Residual-MPPI in the challenging racing game, Gran Turismo Sport (GTS), in which we successfully customize the driving strategy of the champion-level racing agent, GT Sophy 1.0 (Wurman et al., 2022), to adhere to additional constraints.

## 2 PRELIMINARIES

In this section, we provide a concise introduction to two techniques that are used in our proposed algorithm: RQL and MPPI, to establish the foundations for the main technical results.

### 2.1 POLICY CUSTOMIZATION AND RESIDUAL Q-LEARNING

We consider a discrete-time MDP problem defined by a tuple $\mathcal{M} = (\mathcal{X}, \mathcal{U}, r, p)$, where $\mathcal{X} \subseteq \mathcal{R}^n$ is a continuous state space, $\mathcal{U} \subseteq \mathcal{R}^m$ is a continuous action space, $r : \mathcal{X} \times \mathcal{U} \to \mathbb{R}$ is the reward function, and $p : \mathcal{X} \times \mathcal{U} \times \mathcal{X} \to [0, \infty)$ is the state transition probability density function. The prior policy $\pi : \mathcal{X} \to \mathcal{U}$ is trained as an optimal maximum-entropy policy to solve this MDP problem.

Meanwhile, the add-on task is specified by an add-on reward function $r_R : \mathcal{X} \times \mathcal{U} \to \mathbb{R}$. The full task becomes a new MDP defined by a tuple $\hat{\mathcal{M}} = (\mathcal{X}, \mathcal{U}, \omega r + r_R, p)$, where the reward is defined as a weighted sum of the basic reward $r$ and the add-on reward $r_R$. The policy customization task is to find a policy that solves this new MDP. Li et al. (2024) proposed the residual Q-learning framework to solve the policy customization task. Given the prior policy $\pi$, RQL is able to find this customized policy without knowledge of the prior reward $r$, which ensures broader applicability across different prior policies obtained through various methods, including those solely imitating demonstrations.

In particular, as shown in their appendix (Li et al., 2024), when we have access to the prior policy $\pi$, finding the maximum-entropy policy solving the full MDP problem $\hat{\mathcal{M}} = (\mathcal{X}, \mathcal{U}, \omega r + r_R, p)$ is equivalent to solving an augmented MDP problem $\mathcal{M}^{\mathrm{aug}} = (\mathcal{X}, \mathcal{U}, \omega' \log \pi(\boldsymbol{u}|\boldsymbol{x}) + r_R, p)$, where $\omega'$ is a hyper-parameter that balances the optimality between original and add-on tasks.

## 2.2 MODEL PREDICTIVE PATH INTEGRAL

Model Predictive Path Integral (MPPI) (Williams et al., 2017) is a sampling-based model predictive control (MPC) algorithm, which approximates the optimal solution of an (infinite-horizon) MDP through receding-horizon planning. MPPI evaluates the control inputs $U = (\boldsymbol{u}_0, \boldsymbol{u}_1, \cdots, \boldsymbol{u}_{T-1})$ with an accumulated reward function $S_{\boldsymbol{x}_0}(U)$ defined by a reward function $r$ and a terminal value estimator $\phi$:

$$S_{\boldsymbol{x}_0}(U) = \sum_{t=0}^{T-1} r(\boldsymbol{x}_t, \boldsymbol{u}_t) + \phi(\boldsymbol{x}_T), \tag{1}$$

where the intermediate state $\boldsymbol{x}_t$ is calculated by recursively applying the transition dynamics on $\boldsymbol{x}_0$.

By applying an addictive noise sequence $\mathcal{E} = (\boldsymbol{\epsilon}_0, \boldsymbol{\epsilon}_1, \ldots, \boldsymbol{\epsilon}_{T-1})$ to a nominal control input sequence $\hat{U}$, MPPI obtains a disturbed control input sequence as $U = \hat{U} + \mathcal{E}$ for subsequent optimization, where $\mathcal{E}$ follows a multivariate Gaussian distribution with its probability density function defined as $p(\mathcal{E}) = \prod_{t=0}^{T-1} ((2\pi)^m |\Sigma|)^{-\frac{1}{2}} \exp\left(-\frac{1}{2} \boldsymbol{\epsilon}_t^T \Sigma^{-1} \boldsymbol{\epsilon}_t\right)$, where $m$ is the dimension of the action space.

As shown by Williams et al. (2017), the optimal action distribution that solves the MDP is

$$q^*(U) = \frac{1}{\eta} \exp\left(\frac{1}{\lambda} S_{\boldsymbol{x}_0}(U)\right) p(\mathcal{E}), \tag{2}$$

where $\eta$ is the normalizing constant and $\lambda$ is a positive scalar variable. MPPI approximates this distribution by assigning an importance sampling weight $\omega(\mathcal{E}^k)$ to each noise sequence $\mathcal{E}^k$ to update the nominal control input sequence:

$$\boldsymbol{u}_t = \hat{\boldsymbol{u}}_t + \sum_{k=1}^{K} \omega(\mathcal{E}^k) \boldsymbol{\epsilon}_t^k, \tag{3}$$

where $K$ is the number of samples, and $\omega(\mathcal{E}^k)$ could be calculated as

$$\omega(\mathcal{E}^k) = \frac{1}{\mu} \left( S_{\boldsymbol{x}_0}(\hat{U} + \mathcal{E}^k) - \frac{\lambda}{2} \sum_{t=0}^{T-1} \hat{\boldsymbol{u}}_t^T \Sigma^{-1} (\hat{\boldsymbol{u}}_t + 2\boldsymbol{\epsilon}_t^k) \right), \tag{4}$$

where $\mu$ is the normalizing constant.

## 3 METHOD

Our objective is to address the policy customization challenge under the online setting for continuous control. We aim to leverage a pre-trained prior policy $\pi$ with a dynamics model $F$, to approximate the optimal solution to the augmented MDP problem $\mathcal{M}^{\mathrm{aug}}$, in an online manner. To address this problem, we propose a novel algorithm, *Residual Model Predictive Path Integral* (Residual-MPPI), which is broadly applicable to any maximum-entropy prior policy with a dynamics model. The proposed algorithm is summarized in Algorithm 1 and Figure 1. In this section, we first establish the theoretical foundation of our approach by verifying the maximum-entropy property of MPPI. We then refine the MPPI method with the derived formulation to approximate the solution of the $\mathcal{M}^{\mathrm{aug}}$. Lastly, we discuss the dynamics learning and fine-tuning method used in implementation.

## 3.1 RESIDUAL-MPPI

MPPI is a widely used sampling-based MPC algorithm that has demonstrated promising results in various continuous control tasks. To achieve online policy customization, i.e., solving the augmented MDP $\mathcal{M}^{\mathrm{aug}}$ efficiently during online execution, we utilize MPPI as the foundation of our algorithm.

---

**Algorithm 1** Residual-MPPI

---

**Input:** Current state $\boldsymbol{x}_0$; **Output:** Action Sequence $\hat{U} = (\hat{\boldsymbol{u}}_0, \hat{\boldsymbol{u}}_1, \cdots, \hat{\boldsymbol{u}}_{T-1})$.

**Require:** System dynamics $F$; Number of samples $K$; Planning horizon $T$; Prior policy $\pi$; Disturbance covariance matrix $\Sigma$; Add-on reward $r_R$; Temperature scalar $\lambda$; Discounted factor $\gamma$

1: **for** $t = 0, \ldots, T-1$ **do**          $\triangleright$ Initialize the action sequence from the prior policy
2:      $\hat{\boldsymbol{u}}_t \leftarrow \arg\max \pi(\boldsymbol{u}_t | \boldsymbol{x}_t)$
3:      $\boldsymbol{x}_{t+1} \leftarrow \boldsymbol{F}(\boldsymbol{x}_t, \hat{\boldsymbol{u}}_t)$
4: **end for**
5: **for** $k = 1, \ldots, K$ **do**          $\triangleright$ Evaluate the sampled action sequence
6:      Sample noise $\mathcal{E}^k = \{\epsilon_0^k, \epsilon_1^k, \cdots, \epsilon_{T-1}^k\}$
7:      **for** $t = 0, \ldots, T-1$ **do**
8:          $\boldsymbol{x}_{t+1} \leftarrow \boldsymbol{F}(\boldsymbol{x}_t, \hat{\boldsymbol{u}}_t + \epsilon_t^k)$
9:          $S(\mathcal{E}^k) + = \gamma^t \times \left( r_R(\boldsymbol{x}_t, \hat{\boldsymbol{u}}_t + \epsilon_t^k) + \omega' \log \pi(\hat{\boldsymbol{u}}_t + \epsilon_t^k | \boldsymbol{x}_t) \right) - \lambda \hat{\boldsymbol{u}}_t^T \Sigma^{-1} \epsilon_t^k$
10:      **end for**
11: **end for**
12: $\beta = \min_k S(\mathcal{E}^k)$          $\triangleright$ Update the action sequence
13: $\eta \leftarrow \Sigma_{k=1}^K \exp\left( \frac{1}{\lambda} \left( S\left(\mathcal{E}^k\right) - \beta \right) \right)$
14: **for** $k = 1, \ldots, K$ **do**
15:      $\omega(\mathcal{E}^k) \leftarrow \frac{1}{\eta} \exp\left( \frac{1}{\lambda} \left( S\left(\mathcal{E}^k\right) - \beta \right) \right)$
16: **end for**
17: **for** $t = 0, \ldots, T-1$ **do**
18:      $\hat{\boldsymbol{u}}_t + = \sum_{k=1}^K \omega(\mathcal{E}^k) \epsilon_t^k$
19: **end for**
20: **return** $U$

---

As shown in Sec. 2.1, RQL requires the planning algorithm, MPPI, to comply with the principle of maximum entropy. Note that this has been shown in Bhardwaj et al. (2020), where this result established the foundation for their Q-learning algorithm integrating MPPI and model-free soft Q-learning. In Residual-MPPI, this result serves as a preliminary step to ensure that MPPI can be employed as an online planner to solve the augmented MDP $\mathcal{M}^{\text{aug}}$ in policy customization. To better serve our purpose, we provide a self-contained and concise notation of this observation in Proposition 1. Its step-by-step breakdown proof can be found in Appendix A.

**Proposition 1.** *Given an MDP defined by $\mathcal{M} = (\mathcal{X}, \mathcal{U}, r, p)$, with a deterministic state transition $p$ defined with respect to a dynamics model $F$ and a discount factor $\gamma = 1$, the distribution of the action sequence $q^*(U)$ at state $\boldsymbol{x}_0$ in horizon $T$, where each action $\boldsymbol{u}_t, t = 0, \cdots, T-1$ is sequentially sampled from the maximum-entropy policy (Haarnoja et al., 2017) with an entropy weight $\alpha$ is*

$$q^*(U) = \frac{1}{Z_{\boldsymbol{x}_0}} \exp\left( \frac{1}{\alpha} \left( \sum_{t=0}^{T-1} r(\boldsymbol{x}_t, \boldsymbol{u}_t) + V^*(\boldsymbol{x}_T) \right) \right), \tag{5}$$

*where $V^*$ is the soft value function (Haarnoja et al., 2017) and $\boldsymbol{x}_t$ is defined recursively from $\boldsymbol{x}_0$ and $U$ through the dynamics model $F$ as $\boldsymbol{x}_{t+1} = F(\boldsymbol{x}_t, \boldsymbol{u}_t), t = 0, \cdots, T-1$.*

If we have $\lambda = \alpha$ and let $V^*$ be the terminal value estimator, the distribution in equation 5 is equivalent to the one in equation 2, that is the optimal distribution that MPPI tries to approximate, but under the condition that the $p(\mathcal{E})$ is a Gaussian distribution with infinite variance, i.e. a uniform distribution. It suggests that MPPI can well approximate the maximum-entropy optimal policy with a discount factor $\gamma$ close to 1 and a large noise variance. We can then derive Residual-MPPI straightforwardly by defining the evaluation function $S_{\boldsymbol{x}_0}(U)$ in MPPI as

$$S_{\boldsymbol{x}_0}^{\text{aug}}(U) = \sum_{t=0}^{T-1} \gamma^t \cdot \left( r_R(\boldsymbol{x}_t, \boldsymbol{u}_t) + \omega' \log \pi(\boldsymbol{u}_t | \boldsymbol{x}_t) \right), \tag{6}$$

to solve the $\mathcal{M}^{\text{aug}}$, therefore approximates the optimal customized policy online.

The performance and sample efficiency of MPPI depends on the approach to initialize the nominal input sequence $\hat{U}$. In policy customization, the prior policy serves as a natural source for initializing the nominal control inputs. As shown at line 1 in Algorithm 1, by recursively applying the

prior policy, we could initialize a nominal trajectory with a tunable exploration noise to construct a Gaussian prior distribution for sampling. During experiments, we found that including the nominal action sequence as a candidate evaluation sequence can effectively increase sampling stability.

## 3.2 DYNAMICS LEARNING AND FINE-TUNING

In scenarios where an effective dynamics model is unavailable in a prior, it is necessary to develop a learned dynamics model. To this end, we established a dynamics training pipeline utilizing the prior policy for data collection. In our implementation, we employed three main techniques to enhance the model's capacity for accurately predicting environmental states, as follows. The pseudocode of the complete Residual-MPPI deployment pipeline can be found in Algorithm 2 in Appendix B.

**Multi-step Error:** The prediction errors of the imperfect dynamics model accumulate over time steps. In the worst case, compounding multi-step errors can grow exponentially (Venkatraman et al., 2015). To ensure the accuracy of the dynamics over the long term, we use a multi-step error $\sum_{t=0}^{T} \gamma^t (s_t - \hat{s}_t)^2$ as the loss for training, where $s_t$ and $\hat{s}_t$ are the ground-truth and prediction.

**Exploration Noise:** The prior policy's behavior sticks around the optimal behavior to the prior task, which means the roll-out samples collected using the prior policy concentrated to a limited region in the state space. It limits the generalization performance of the dynamics model under the policy customization setting. Therefore, during the sampling process, we add Gaussian exploration noises to the prior policy actions to enhance sample diversity.

**Fine-tuning with Online Data:** Since the residual-MPPI planner solves the customized task objective, its behavior is different from the prior. Thus, the planner may encounter states not contained in the dynamics training dataset collected by prior policy, on which the learned dynamics could be inaccurate. In this case, we can iteratively collect data with Residual-MPPI and update the dynamics model using the in-distribution data. We refer to the variant with online dynamics fine-tuning as *few-shot Residual-MPPI* and the variant without fine-tuning as *zero-shot Residual-MPPI*.

## 4 MuJoCo EXPERIMENTS

In this section, we evaluate the performance of the proposed algorithms in different environments selected from MuJoCo (Todorov et al., 2012). In Sec. 4.1, we provide the configurations of our experiments, including the settings of policy customization tasks in different environments, baselines, and evaluation metrics. In Sec. 4.2, we present and analyze the experimental results. Please refer to the appendices for detailed experiment configurations, implementations, and visualizations.

### 4.1 EXPERIMENT SETUP

**Environments.** In each environment, we design the add-on rewards to illustrate a customized specifications such as behavior preference or additional task objectives in the practical application scenarios. In HalfCheetah and Swimmer, we apply an add-on penalty on the angle of a certain joint, which is a common issue in deployment if the corresponding motor is broken; In Hopper, we apply an extra reward on height; In Ant, we apply an add-on reward for moving along the y-axis. The configurations of environments and training parameters can be found in Appendix C.

**Baselines.** In our experiments, we compare the performance of the proposed residual-MPPI algorithm against seven baselines, including the prior policy, four alternative MPPI variants, and two RL-based baselines. Note that except for Greedy-MPPI, the remaining MPPI baselines have access to the underlying reward or value function of the prior policy. These baselines show that Residual-MPPI is still the ideal choice, even with privileged access to additional reward or value information.

- **Prior Policy:** We utilize SAC to train the prior policy on the basic task for policy customization. At test time, we evaluate its performance on the overall task without customization, which serves as a baseline to show the effectiveness of policy customization.
- **Greedy-MPPI:** Firstly, we introduce Greedy-MPPI, which samples the action sequences from the prior policy but only optimizes the control actions for the add-on reward $r_R$, i.e., removing the $\log \pi$ reward term of the proposed Residual-MPPI. Through comparison with Greedy-MPPI, we aim to show the necessity and effect of including $\log \pi$ as a reward in Residual-MPPI.

- **Full-MPPI:** Next, we apply the MPPI with no prior on the MDP of the full task (i.e., $\omega r + r_R$). We aim to compare Residual-MPPI against it to validate that our proposed algorithm can effectively leverage the prior policy to boost online planning.

- **Guided-MPPI:** Furthermore, we introduce Guided-MPPI, which samples the control actions from the prior policy and also has privileged access to the full reward information, i.e., $\omega r + r_R$. By comparing it against Guided-MPPI, we aim to show the advantages and the necessity of Residual-MPPI even with granted access to the prior reward.

- **Valued-MPPI:** While it is not compatible with our problem setting, we further consider the case where the value function of the prior policy is accessible. We construct a variant of Guided-MPPI with this value function as a terminal value estimator like Argenson & Dulac-Arnold (2020).

- **Residual-SAC:** We also include the Residual-SAC as a RL-based baseline to set an upper bound performance of the policy customization task without knowing the prior reward. We used the final checkpoint (4M steps) as well as checkpoints with an equivalent amount of data to dynamics training (200K steps) to illustrate the superior sample efficiency of Residual-MPPI.

- **Fulltask-SAC:** Finally, we report the SAC policy trained on the full task as a reference to set an upper bound for the policy customization task's performance.

**Metric.** We aim to validate a policy's performance with total reward, i.e., $\omega r + r_R$. When the add-on task contradicts the basic task, the optimal policy may trade off its performance on the basic task to maximize the total reward. Therefore, we further measure the basic and add-on rewards separately to monitor the trade-off achieved by the customized policy, i.e., how well the performance on the basic task is maintained and how much the policy is optimized toward the add-on task.

## 4.2 RESULTS AND DISCUSSIONS

The experimental results, summarized in Table 1, demonstrate the effectiveness of Residual-MPPI for online policy customization. The ablation results of the planning parameters in MuJoCo and visualization can be found in Appendix F.2 and Appendix E.1. Also, we conduct experiments upon the same MuJoCo environments with IL prior policies, whose results are summarized in Table 10 in Appendix F.1. Across the tested environments, the customized policies achieve significant improvements over the prior policies in meeting the objectives of the add-on tasks while maintaining a similar level of performance in terms of the basic task objectives. aIn particular, we observe three key advantages of Residual-MPPI over baseline approaches.

*First, Residual-MPPI demonstrates significantly higher sample-efficiency over the RL-based baselines.* While Fulltask-SAC and Residual-SAC achieve better performance than planning-based approaches through extensive online exploration (4M steps), our focus is on online policy customization. Thus, we consider their performance as upper bounds in our online setting. With a substantially smaller dataset for dynamics model training (2K steps), Residual-MPPI achieves total rewards comparable to Residual-SAC in the HalfCheetah, Hopper, and Ant environments. In contrast, Residual-SAC performs significantly worse when trained on the same limited amount of data. *Second, Residual-MPPI can strike a better trade-off between the basic and add-on rewards than Greedy-MPPI.* Greedy-MPPI optimizes the objective of a biased MDP $\mathcal{M}^{\text{add}} = (\mathcal{X}, \mathcal{U}, \mathbf{r_R}, \mathbf{p})$. It relies solely on sampling from the prior policy to heuristically regularize the optimized action sequences to remain close to the prior policy. While this heuristic approach helps maintain performance on the basic task objective, it is prone to suboptimality in tasks where the add-on reward is sparse or orthogonal to the basic reward. For example, in the Ant environment, the agent is rewarded for making progress along the x-axis in the basic task, while the add-on reward encourages progress along the y-axis. Since these reward terms are orthogonal, exclusively optimizing for progress along the y-axis compromises the progress along the x-axis, leading to a noticeable performance gap for Greedy-MPPI. This limitation is further amplified in the more complex racing task, as we will demonstrate in the next section.

*Finally, Residual-MPPI outperforms MPPI variants that have privileged access to the prior policy's reward or value functions.* Despite having full reward knowledge, Full-MPPI fails to complete the task. Without sampling control actions from the prior policy, Full-MPPI suffers from poor sample efficiency and achieves subpar performance. Guided-MPPI can achieve better performance, given its access to action samples from the prior policy. However, it still underperforms Residual-MPPI

Table 1: Experimental Results of Zero-shot Residual-MPPI in MuJoCo

| Env. | Policy | Full Task | Basic Task | Add-on Task | |
|---|---|---|---|---|---|
| | | Total Reward | Basic Reward | $|\bar{\theta}|$ | Add-on Reward |
| Half Cheetah | Prior Policy | $1000.7 \pm 88.8$ | $2449.8 \pm 52.3$ | $0.14 \pm 0.00$ | $-1449.1 \pm 45.3$ |
| | Greedy-MPPI | $\mathbf{1939.9 \pm 134.7}$ | $2180.9 \pm 87.3$ | $\mathbf{0.02 \pm 0.01}$ | $\mathbf{-241.0 \pm 50.3}$ |
| | Full-MPPI | $-3595.1 \pm 322.7$ | $-1167.3 \pm 144.0$ | $0.24 \pm 0.03$ | $-2427.7 \pm 320.3$ |
| | Guided-MPPI | $1849.6 \pm 151.0$ | $2154.6 \pm 95.7$ | $0.03 \pm 0.01$ | $-305.0 \pm 58.7$ |
| | Valued-MPPI | $1760.7 \pm 478.8$ | $\mathbf{2201.8 \pm 258.3}$ | $0.04 \pm 0.02$ | $-441.0 \pm 222.5$ |
| | Residual-MPPI | $\mathbf{1936.2 \pm 109.3}$ | $2178.6 \pm 71.9$ | $\mathbf{0.02 \pm 0.00}$ | $\mathbf{-242.3 \pm 40.5}$ |
| | Residual-SAC (200K) | $-265.0 \pm 919.0$ | $455.4 \pm 678.6$ | $0.07 \pm 0.03$ | $720.4 \pm 251.8$ |
| | Residual-SAC (4M) | $2184.5 \pm 29.7$ | $2233.7 \pm 29.3$ | $0.00 \pm 0.00$ | $-49.2 \pm 1.7$ |
| | Fulltask-SAC | $2149.9 \pm 28.6$ | $2214.5 \pm 27.2$ | $0.01 \pm 0.00$ | $-64.5 \pm 2.4$ |

| Env | Policy | Total Reward | Basic Reward | $|\bar{\theta}|$ | Add-on Reward |
|---|---|---|---|---|---|
| Swimmer | Prior Policy | $-245.2 \pm 5.6$ | $345.8 \pm 3.2$ | $0.59 \pm 0.01$ | $-591.0 \pm 5.8$ |
| | Greedy-MPPI | $\mathbf{-58.9 \pm 5.4}$ | $275.8 \pm 3.1$ | $\mathbf{0.33 \pm 0.01}$ | $\mathbf{-334.7 \pm 7.4}$ |
| | Full-MPPI | $-1686.6 \pm 106.7$ | $14.1 \pm 6.3$ | $1.70 \pm 0.11$ | $-1700.7 \pm 106.2$ |
| | Guided-MPPI | $-149.0 \pm 5.6$ | $292.9 \pm 3.8$ | $0.44 \pm 0.01$ | $-441.9 \pm 7.2$ |
| | Valued-MPPI | $-205.8 \pm 6.3$ | $\mathbf{335.1 \pm 1.6}$ | $0.54 \pm 0.01$ | $-540.9 \pm 6.3$ |
| | Residual-MPPI | $\mathbf{-60.0 \pm 5.2}$ | $275.8 \pm 3.4$ | $\mathbf{0.34 \pm 0.01}$ | $\mathbf{-335.9 \pm 7.6}$ |
| | Residual-SAC (200K) | $-209.0 \pm 67.6$ | $2.1 \pm 15.5$ | $0.21 \pm 0.07$ | $-221.1 \pm 72.7$ |
| | Residual-SAC (4M) | $-10.5 \pm 24.1$ | $-1.5 \pm 16.9$ | $0.01 \pm 0.02$ | $-9.0 \pm 16.6$ |
| | Fulltask-SAC | $-4.2 \pm 17.1$ | $2.1 \pm 17.6$ | $0.01 \pm 0.00$ | $-6.3 \pm 3.0$ |

| Env. | Policy | Total Reward | Basic Reward | $\bar{z}$ | Add-on Reward |
|---|---|---|---|---|---|
| Hopper | Prior Policy | $7252.7 \pm 49.2$ | $3574.5 \pm 9.7$ | $1.37 \pm 0.00$ | $3678.2 \pm 48.3$ |
| | Greedy-MPPI | $\mathbf{7367.0 \pm 199.4}$ | $3553.0 \pm 58.4$ | $\mathbf{1.38 \pm 0.01}$ | $3814.0 \pm 156.8$ |
| | Full-MPPI | $20.5 \pm 3.0$ | $3.6 \pm 0.7$ | $1.24 \pm 0.00$ | $16.9 \pm 2.4$ |
| | Guided-MPPI | $6121.3 \pm 1590.1$ | $3067.8 \pm 679.0$ | $1.35 \pm 0.03$ | $3053.4 \pm 917.7$ |
| | Valued-MPPI | $7243.9 \pm 75.7$ | $\mathbf{3562.7 \pm 14.5}$ | $1.37 \pm 0.01$ | $3681.2 \pm 74.6$ |
| | Residual-MPPI | $\mathbf{7363.0 \pm 254.9}$ | $3547.6 \pm 78.0$ | $\mathbf{1.38 \pm 0.01}$ | $\mathbf{3815.4 \pm 186.4}$ |
| | Residual-SAC (200K) | $3543.1 \pm 478.9$ | $1019.8 \pm 94.3$ | $1.27 \pm 0.01$ | $2523.2 \pm 405.5$ |
| | Residual-SAC (4M) | $7682.5 \pm 178.2$ | $2310.4 \pm 106.8$ | $1.54 \pm 0.01$ | $5372.0 \pm 75.8$ |
| | Fulltask-SAC | $7825.3 \pm 36.9$ | $2934.5 \pm 27.6$ | $1.49 \pm 0.00$ | $4890.8 \pm 39.6$ |

| Env | Policy | Total Reward | Basic Reward | $\bar{v}_y$ | Add-on Reward |
|---|---|---|---|---|---|
| Ant | Prior Policy | $6333.7 \pm 753.9$ | $6177.1 \pm 703.7$ | $0.16 \pm 0.22$ | $156.6 \pm 200.5$ |
| | Greedy-MPPI | $6104.2 \pm 1532.0$ | $5092.8 \pm 1305.2$ | $\mathbf{1.01 \pm 0.27}$ | $\mathbf{1011.3 \pm 277.7}$ |
| | Full-MPPI | $-2767.7 \pm 154.0$ | $-2764.4 \pm 114.2$ | $-0.00 \pm 0.11$ | $-3.3 \pm 108.0$ |
| | Guided-MPPI | $5160.9 \pm 1963.0$ | $4999.8 \pm 1887.9$ | $0.16 \pm 0.22$ | $161.2 \pm 217.7$ |
| | Valued-MPPI | $6437.0 \pm 1021.9$ | $\mathbf{6230.7 \pm 959.0}$ | $0.21 \pm 0.20$ | $206.3 \pm 196.3$ |
| | Residual-MPPI | $\mathbf{6846.7 \pm 647.8}$ | $5984.8 \pm 541.5$ | $\mathbf{0.86 \pm 0.19}$ | $\mathbf{861.8 \pm 189.8}$ |
| | Residual-SAC (200K) | $-1175.5 \pm 157.3$ | $-1178.3 \pm 156.4$ | $0.00 \pm 0.00$ | $2.7 \pm 3.9$ |
| | Residual-SAC (4M) | $6962.9 \pm 342.9$ | $5710.2 \pm 252.0$ | $1.25 \pm 0.13$ | $1252.7 \pm 127.3$ |
| | Fulltask-SAC | $7408.6 \pm 312.0$ | $3100.3 \pm 184.4$ | $4.31 \pm 0.21$ | $4308.3 \pm 209.2$ |

The evaluation results are in the form of mean $\pm$ std over the 500 running episodes. The total reward is calculated on full task, whose reward is $\omega r + r_R$.

across all tasks, as it is constrained by its limited ability to reason the long-term effects of actions within its finite planning horizons. Conversely, Residual-MPPI implicitly incorporates the original task reward through the prior policy log-likelihood. The $\log \pi$ term also serves as a proxy of the prior policy's Q-function, informing the planner of the impact of the action sequences beyond the planning horizon. While Valued-MPPI addresses long-term impacts by incorporating the prior Q-function, it struggles to fully align with the complete task reward. Valued-MPPI considers the long-term impact by incorporating the prior Q-function but fails to match the full task reward setting, resulting in better performance on the basic task but suboptimum on the full task.

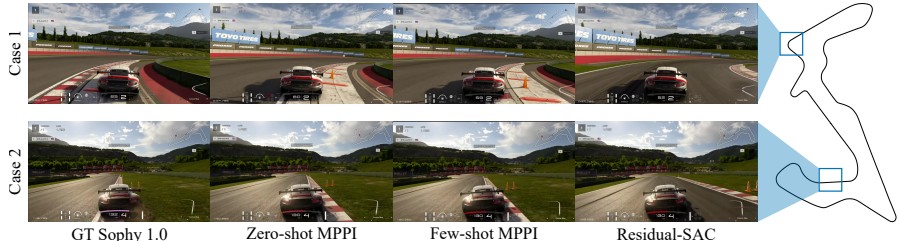

Figure 2: In-game screen shots of Policy Behavior on Different Road Sections

## 5 CUSTOMIZING CHAMPION-LEVEL AUTONOMOUS RACING POLICY

With the effective results in standard benchmark environments, we are further interested in whether the proposed algorithm can be applied to effectively customize an advanced policy for a more sophisticated continuous control task. Gran Turismo Sport (GTS), a realistic racing simulator that simulates high-fidelity racing dynamics, complex environments at an independent, fixed frequency with realistic latency as in real-world control systems, serves as an ideal test bed. GT Sophy 1.0 (Wurman et al., 2022), a DRL-based racing policy, has been shown to outperform the world's best drivers among the global GTS player community. To further investigate the scalability of our algorithm and its robustness when dealing with complex environments and advanced policies, we carried out further experiments on GT Sophy 1.0 in the GTS environment.

### 5.1 EXPERIMENT SETUP

Though GT Sophy 1.0 is a highly advanced system with superhuman racing skills, it tends to exhibit too aggressive route selection as the well-trained policy has the ability to keep stable on the simulated grass. However, in real-world racing, such behaviors would be considered dangerous and fragile because of the time-varying physical properties of real grass surfaces and tires. Therefore, we establish the task as customizing the policy to a safer route selection. Formally, the customization goal is to help the GT Sophy 1.0 drive *on course* while maintaining its superior racing performance. This kind of customization objective could potentially be used to foster robust sim-to-real transfer of agile control policies for many related problem domains.

We adopt a simple MLP architecture to design a dynamics model and train it using the techniques introduced in Sec. 3.2. The configurations of the environments and implementation details can be found in Appendix D. As the pre-trained GT Sophy 1.0 policy is constructed with complex rewards and regulations, we adopt the average lap time and number of off-course steps as the metrics. In addition to zero-shot Residual-MPPI, we also evaluate a few-shot version of our algorithm, in which we iteratively update the dynamics with the customized planner's trajectories to further improve the performance. We also consider Residual-SAC (Li et al., 2024) as another baseline to validate the advantage of online Residual-MPPI against RL-based solutions in the challenging racing problem. Notably, we are only given access to the policy network of GT Sophy 1.0, which prevents us from evaluating Valued-MPPI, as it requires access to the critic function. Therefore, we only include Greedy-MPPI and Guided-MPPI as MPPI variants for comparison in the GTS environment.

### 5.2 RESULTS AND DISCUSSIONS

The experimental results, summarized in Table 2, demonstrate that Residual-MPPI significantly enhances the safety of GT Sophy 1.0 by reducing its off-course steps, albeit with a marginal increase in lap time. Further improvements are observed after employing data gathered during the customization process to fine-tune the dynamics under a few-shot setting. This few-shot version of Residual-MPPI outperforms the zero-shot version in terms of lap time and off-course steps.

The ablation results of the planning parameters and visualization in GTS can be found in Appendix F.3 and Appendix E.2, which clearly demonstrates the effectiveness of the proposed method by greatly reducing the off-course steps. In the detailed route selection visualization, it can be observed that Few-shot MPPI chose a safer and faster racing line compared to Zero-shot MPPI with

Table 2: Experimental Results of Residual-MPPI in GTS

| Policy | GT Sophy 1.0 | Zero-shot MPPI | Few-shot MPPI | Residual-SAC (80K laps) |
|---|---|---|---|---|
| Lap Time | $117.77 \pm 0.08$ | $123.34 \pm 0.22$ | $122.93 \pm 0.14$ | $130.00 \pm 0.13$ |
| Off-course Steps | $93.13 \pm 1.98$ | $9.03 \pm 3.33$ | $4.43 \pm 2.39$ | $0.87 \pm 0.78$ |
| Policy | Full-MPPI | Guided-MPPI | Greedy-MPPI | Residual-SAC (2K laps) |
| Lap Time | *Failed | *Failed | *Failed | *Failed |
| Off-course Steps | *Failed | *Failed | *Failed | *Failed |

The evaluation results are in the form of mean ± std over 30 laps. *Failed baseline is not able to finish a complete lap. Valued-MPPI is not available since we only have access to the policy network of GT Sophy 1.0.

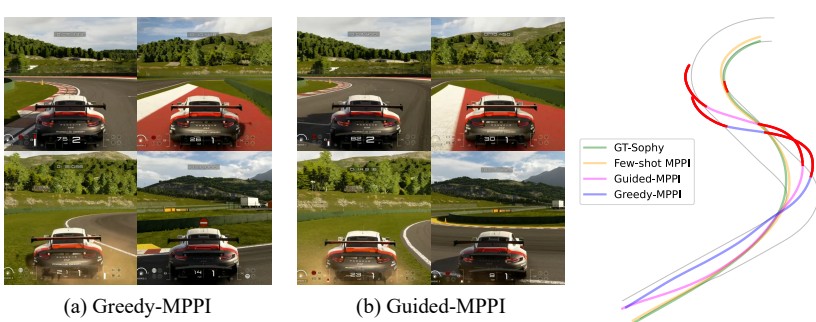

(a) Greedy-MPPI  (b) Guided-MPPI

Figure 3: Guided-MPPI and Greedy-MPPI Results in GTS. (a) In-game screenshots of Greedy-MPPI; (b) In-game screenshots of Guided-MPPI. Red parts indicate off-course behaviors. Both baselines cannot drive the vehicle effectively, completely going off track at the first corner.

more accurate dynamics. Though the customized policies can not eliminate all the off-course steps, it is worth noting that these violations are minor (i.e., slightly touching the course boundaries) compared to GT Sophy 1.0, making the customized policies safe enough to stay on course.

**Residual-SAC**, compared with the Residual-MPPI, yields a very conservative customized policy. As shown in Figure 2, it is obvious that the Residual-SAC policy behaves sub-optimally and overly yields to the on-course constraints. It is worth noting that over *80,000 laps* of roll-outs are collected to achieve the current performance of Residual-SAC. In contrast, the data used to train GTS dynamics for Residual-MPPI, along with the online data for dynamics model fine-tuning, amounted to only approximately *2,000* and *100* laps, respectively. As discussed in safe RL and curriculum learning methods (Mo et al., 2021; Anzalone et al., 2022), when training with constraints, it is easy for RL to yield to an overly conservative behavior. At the checkpoint with 2k laps of training data, Residual-SAC could not finish a lap yet, which highlights the outstanding data-efficiency of Residual-MPPI.

**Guided-MPPI**, however, cannot finish the track stably, as shown in figure 3. Similar to what we have analyzed in the MuJoCo experiments, Guided-MPPI suffers from its limited ability to account for the long-term impacts of actions given the finite planning horizon. This limitation leads to more severe consequences in complex tasks that require long-term, high-level decision-making, such as route selection during racing, which leads to failure in GTS.

**Greedy-MPPI**, as mentioned above, also leads to severe failure in the complex GTS task, further emphasizing the importance of the $\log \pi$ term in Residual-MPPI's objective function. In GTS, the Greedy-MPPI only optimizes the trajectories by not allowing the policy to be off-course, where a straightforward local optimal but undesirable solution would be just staying still. Such a trivial policy would behave better on the add-on tasks (stay on course) but not on the full task (stay on course while maintaining racing superiority). From the theoretical perspective, the significance of the $\log \pi$ term goes far beyond a simple regularization term. It is the key factor in addressing the policy customization problem, which is inherently a part of the joint optimization objective, encoding and passing the information of the original reward in a theoretically sound manner.

## 6   RELATED WORKS

**Model-based RL.** Many works focused on combining learning-based agents with classical planning to enhance performance, especially in various model-based RL approaches. MuZero (Schrittwieser et al., 2020) employs a learned Actor as the search policy during the MCTS process and utilizes the critic as the terminal value at the leaf nodes; TD-MPC2 (Hansen et al., 2023) follows a similar approach and extends it to continuous task by utilizing MPPI as the continuous planner. RL-driven MPPI (Qu et al., 2024) chooses the distributional SACs as the prior policy and also adopts the MPPI as the planner. However, those methods mainly address the planning within the same task and have a full reward and value function, while policy customization requires jointly optimizing an add-on reward and the underlying reward of the prior policy, which is unknown in a priori. In general cases, the prior policy may not provide a critic, as is the case with algorithms like policy gradient (Shi et al., 2019). Moreover, the value function will change with the new reward, making it infeasible to be applied. Therefore, these methods are not able to solve policy customization tasks directly.

**Policy Adaptation.** There have been numerous works considering on the topic of policy adaptation. Some works require an extra demonstration dataset to adapt the policy to the new task, like Ceil (Liu et al., 2023) and Prompt-DT (Xu et al., 2022). However, such expert demonstrations are unavailable in the online policy customization settings. Some works focus on adaptation via RL fine-tuning: JSRL (Uchendu et al., 2023) uses a prior policy as the guiding policy to form a curriculum for the exploration policy; AWAC (Nair et al., 2020) combines sample-efficient dynamic programming with maximum likelihood policy updates to leverage large amounts of offline data for quickly performing online fine-tuning of RL policies. However, the main objective in RL fine-tuning is to maximize the policy return on the new task. In contrast, policy customization (Li et al., 2024) aims to find a policy that can maintain the prior performance on the basic task at the same time.

**Planning with Prior Policy.** Some works address a similar problem of customizing pre-trained prior policies towards additional requirements. Efficient game-theoretic planning (Li et al., 2022) uses a planner to construct human behavior within the pre-trained prediction model. Deep Imitative Model (Rhinehart et al., 2018) aims to direct an imitative policy to arbitrary goals during online usage. However, they require full reward modeling with hand-designed features, or the add-on task is in the form of specific goals. Residual-MCTS proposed in RQL (Li et al., 2024) solves the policy customization online via planning but is limited to tasks with discrete action spaces. MBOP (Argenson & Dulac-Arnold, 2020) solves the offline RL problem through planning by learning dynamics, action prior, and values from an offline dataset. LOOP (Sikchi et al., 2022) focused on learning off-policy with samples from an online planner. However, both approaches require knowledge of the basic reward function and thus cannot be directly applied to our setting.

## 7   LIMITATIONS AND FUTURE WORK

In this work, we propose *Residual-MPPI*, which integrates RQL into the MPPI framework, resulting in an online planning algorithm that customizes the prior policy at the execution time. By conducting experiments in MuJoCo and GTS, we show the effectiveness of our methods against the baselines. Here, we discuss some limitations of the current method, inspiring several future research directions.

**Terminal Value Estimation.** Similar to most online planning methods, our algorithm also needs a sufficiently long horizon to reduce the error brought by the absence of a terminal value estimator. Though utilizing prior policy partially addresses the problem discussed in Sec. 4.2, it can not eliminate the error entirely. Techniques like JSRL (Uchendu et al., 2023) and RL-driven MPPI (Qu et al., 2024) could be incorporated in the proposed framework in the few-shot setting by using the collected data to learn a Residual Q-function online as a terminal value estimator.

**Sim2Real Transferability.** Sim2real will still face several additional challenges, which mainly comes from the suboptimal prior policy and the domain gap. Prior policies without careful modeling or sufficient training could mislead the evaluation step of the proposed algorithm and result in poor planning. Also, the proposed algorithm relies on an accurate dynamics model to correctly roll out. In the future, we can introduce more advanced methods in policies and dynamics training, such as diffusion policies (Chi et al., 2023) and world models (Micheli et al., 2022), to improve the prior policies and dynamics. We look forward to further extending residual-MPPI to achieve sim-to-sim and sim-to-real for challenging agile, continuous control tasks by addressing these limitations.

ACKNOWLEDGMENT

The authors thank Ce Hao for valuable discussions and kind help. This work was supported by Sony AI, and Polyphony Digital Inc., which provided the Gran Turismo Sport framework.

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

# A   DERIVATION

In this section, we provide a detailed proof of Proposition 1. We start with introducing a lemma to expand the action distribution $q^*(U)$.

**Lemma 1.** *Given an MDP with a deterministic state transition $p$, the distribution of the action sequence $q(U)$ at state $\boldsymbol{x}_0$ in horizon $T$, where each action $\boldsymbol{u}_t, t = 0, \cdots, T-1$ is sequentially sampled from a policy $\pi(\boldsymbol{u}_t|\boldsymbol{x}_t)$ is*

$$q(U) = \prod_{t=0}^{T-1} \pi(\boldsymbol{u}_t|\boldsymbol{x}_t). \tag{7}$$

*Proof.* Let $\pi(\boldsymbol{u}_0, \boldsymbol{u}_1, \cdots, \boldsymbol{u}_{T-1}|\boldsymbol{x}_0)$ denote the expanded notation of $q(U)$ by substituting $u_t$ and maximum-entropy policy $\pi$. Firstly, we apply the conditional probability formula to expand the equation:

$$\pi(\boldsymbol{u}_0, \boldsymbol{u}_1, \cdots, \boldsymbol{u}_{T-1}|\boldsymbol{x}_0) = \pi(\boldsymbol{u}_0|\boldsymbol{x}_0) \int_{\mathcal{X}} \pi(\boldsymbol{u}_1, \cdots, \boldsymbol{u}_{T-1}|\boldsymbol{x}_0, \boldsymbol{u}_0, \boldsymbol{x}_1')p(\boldsymbol{x}_1'|\boldsymbol{x}_0, \boldsymbol{u}_0)d\boldsymbol{x}_1'. \tag{8}$$

Considering the Markov property of the problem,

$$\pi(\boldsymbol{u}_k, \cdots, \boldsymbol{u}_{k+N}|\boldsymbol{x}_0, \boldsymbol{x}_1', \cdots, \boldsymbol{x}_{k-1}', \boldsymbol{u}_0, \boldsymbol{u}_{k-1}, \boldsymbol{x}_k') = \pi(\boldsymbol{u}_k, \cdots, \boldsymbol{u}_{k+N}|\boldsymbol{x}_k'), \tag{9}$$

Eq. equation 8 could be further expanded recursively till the end:

$$\begin{aligned}
\pi(\boldsymbol{u}_0, \boldsymbol{u}_1, \cdots, \boldsymbol{u}_{T-1}|\boldsymbol{x}_0) &= \pi(\boldsymbol{u}_0|\boldsymbol{x}_0) \int_{\mathcal{X}} \pi(\boldsymbol{u}_1, \cdots, \boldsymbol{u}_{T-1}|\boldsymbol{x}_0, \boldsymbol{u}_0, \boldsymbol{x}_1')p(\boldsymbol{x}_1'|\boldsymbol{x}_0, \boldsymbol{u}_0)d\boldsymbol{x}_1'. \\
&= \pi(\boldsymbol{u}_0|\boldsymbol{x}_0) \int_{\mathcal{X}} \pi(\boldsymbol{u}_1, \cdots, \boldsymbol{u}_{T-1}|\boldsymbol{x}_1')p(\boldsymbol{x}_1'|\boldsymbol{x}_0, \boldsymbol{u}_0)d\boldsymbol{x}_1' \\
&= \pi(\boldsymbol{u}_0|\boldsymbol{x}_0) \int_{\mathcal{X}} \pi(\boldsymbol{u}_1|\boldsymbol{x}_1')p(\boldsymbol{x}_1'|\boldsymbol{x}_0, \boldsymbol{u}_0) \\
&\qquad \int_{\mathcal{X}} \pi(\boldsymbol{u}_2, \cdots, \boldsymbol{u}_{T-1}|\boldsymbol{x}_1')p(\boldsymbol{x}_2'|\boldsymbol{x}_1, \boldsymbol{u}_1)d\boldsymbol{x}_1'd\boldsymbol{x}_2' \\
&= \cdots \\
&= \pi(\boldsymbol{u}_0|\boldsymbol{x}_0) \int_{\mathcal{X}} p(\boldsymbol{x}_1'|\boldsymbol{x}_0, \boldsymbol{u}_0)\pi(\boldsymbol{u}_1|\boldsymbol{x}_1') \\
&\qquad \int \cdots \int_{\mathcal{X}} \prod_{t=2}^{T-1} \pi(\boldsymbol{u}_t|\boldsymbol{x}_t')p(\boldsymbol{x}_t'|\boldsymbol{x}_{t-1}', \boldsymbol{u}_{t-1})d\boldsymbol{x}_1' \cdots d\boldsymbol{x}_{T-1}'.
\end{aligned} \tag{10}$$

Since the state transition $p$ is a deterministic function

$$p\left(\boldsymbol{x}_{t+1}'|\boldsymbol{x}_t, \boldsymbol{u}_t\right) = \delta(\boldsymbol{x}_{t+1}', F(\boldsymbol{x}_t, \boldsymbol{u}_t)), \tag{11}$$

the integral signs could be eliminated by defining $\boldsymbol{x}_{t+1} = F(\boldsymbol{x}_t, \boldsymbol{u}_t)$:

$$
\begin{aligned}
\pi(\boldsymbol{u}_0, \boldsymbol{u}_1, \cdots, \boldsymbol{u}_{T-1}|\boldsymbol{x}_0) = {}& \pi(\boldsymbol{u}_0|\boldsymbol{x}_0) \int_{\mathcal{X}} p(\boldsymbol{x}_1'|\boldsymbol{x}_0, \boldsymbol{u}_0) \pi(\boldsymbol{u}_1|\boldsymbol{x}_1') \\
& \int \cdots \int_{\mathcal{X}} \prod_{t=2}^{T-1} \pi(\boldsymbol{u}_t|\boldsymbol{x}_t') p(\boldsymbol{x}_t'|\boldsymbol{x}_{t-1}', \boldsymbol{u}_{t-1}) d\boldsymbol{x}_1' \cdots d\boldsymbol{x}_{T-1}' \\
= {}& \pi(\boldsymbol{u}_0|\boldsymbol{x}_0) \int_{\mathcal{X}} \delta\left(\boldsymbol{x}_{t+1}', F(\boldsymbol{x}_t, \boldsymbol{u}_t)\right) \pi(\boldsymbol{u}_1|\boldsymbol{x}_1') \\
& \int \cdots \int_{\mathcal{X}} \prod_{t=2}^{T-1} \pi(\boldsymbol{u}_t|\boldsymbol{x}_t') p(\boldsymbol{x}_t'|\boldsymbol{x}_{t-1}', \boldsymbol{u}_{t-1}) d\boldsymbol{x}_1' \cdots d\boldsymbol{x}_{T-1}' \\
= {}& \pi(\boldsymbol{u}_0|\boldsymbol{x}_0) \pi(\boldsymbol{u}_1|\boldsymbol{x}_1) \int_{\mathcal{X}} p(\boldsymbol{x}_2'|\boldsymbol{x}_1, \boldsymbol{u}_1) \pi(\boldsymbol{u}_2|\boldsymbol{x}_2') \\
& \int \cdots \int_{\mathcal{X}} \prod_{t=3}^{T-1} \pi(\boldsymbol{u}_t|\boldsymbol{x}_t') p(\boldsymbol{x}_t'|\boldsymbol{x}_{t-1}', \boldsymbol{u}_{t-1}) d\boldsymbol{x}_2' \cdots d\boldsymbol{x}_{T-1}' \\
= {}& \cdots \\
= {}& \prod_{t=0}^{T-1} \pi(\boldsymbol{u}_t|\boldsymbol{x}_t).
\end{aligned}
\tag{12}
$$

QED

With Lemma 1, now we provide a detailed proof of Theorem 1.

**Theorem 1.** *Given an MDP defined by $\mathcal{M} = (\mathcal{X}, \mathcal{U}, r, p)$, with a deterministic state transition $p$ defined with respect to a dynamics model $F$ and a discount factor $\gamma = 1$, the distribution of the action sequence $q^*(U)$ at state $\boldsymbol{x}_0$ in horizon $T$, where each action $\boldsymbol{u}_t, t = 0, \cdots, T-1$ is sequentially sampled from the maximum-entropy policy (Haarnoja et al., 2017) with an entropy weight $\alpha$ is*

$$
q^*(U) = \frac{1}{Z_{\boldsymbol{x}_0}} \exp\left(\frac{1}{\alpha}\left(\sum_{t=0}^{T-1} r(\boldsymbol{x}_t, \boldsymbol{u}_t) + V^*(\boldsymbol{x}_T)\right)\right),
\tag{13}
$$

*where $V^*$ is the soft value function (Haarnoja et al., 2017) and $\boldsymbol{x}_t$ is defined recursively from $\boldsymbol{x}_0$ and $U$ through the dynamics model $F$ as $\boldsymbol{x}_{t+1} = F(\boldsymbol{x}_t, \boldsymbol{u}_t), t = 0, \cdots, T-1$.*

*Proof.* The maximum-entropy policy with an entropy weight $\alpha$ solves the problem $\mathcal{M}$ following the one-step Boltzmann distribution:

$$
\pi^*(\boldsymbol{u}_t|\boldsymbol{x}_t) = \frac{1}{Z_{\boldsymbol{x}_t}} \exp\left(\frac{1}{\alpha} Q^*(\boldsymbol{x}_t, \boldsymbol{u}_t)\right),
\tag{14}
$$

where $Z_{\boldsymbol{x}_t}$ is the normalization factor defined as $\int_{\mathcal{U}} \exp\left(\frac{1}{\alpha} Q^*(\boldsymbol{x}_t, \boldsymbol{u}_t)\right) d\boldsymbol{u}_t$ and $Q^*(\boldsymbol{x}_t, \boldsymbol{u}_t)$ is the soft Q-function as defined in (Haarnoja et al., 2017):

$$
Q^*(\boldsymbol{x}_t, \boldsymbol{u}_t) = r(\boldsymbol{x}_t, \boldsymbol{u}_t) + \alpha \log \int_{\mathcal{U}} \exp\left(\frac{1}{\alpha} Q^*(\boldsymbol{x}_{t+1}, \boldsymbol{u}_{t+1})\right) d\boldsymbol{u}.
\tag{15}
$$

With Lemma 1, the optimal action distribution could be rewritten as:

$$
q^*(U) = \prod_{t=0}^{T-1} \pi^*(\boldsymbol{u}_t|\boldsymbol{x}_t)
\tag{16}
$$

where each $\boldsymbol{x}_t$ is defined recursively from $\boldsymbol{x}_0$ and $U$ through the dynamics model $F$ as $\boldsymbol{x}_{t+1} = F(\boldsymbol{x}_t, \boldsymbol{u}_t), t = 0, \cdots, T-1$. By substituting Eq. equation 14 and Eq. equation 15, Eq. equation 16 could be further expanded:

$$q^*(U) \tag{17a}$$

$$= \frac{1}{\prod_{t=0}^{T-1} Z_{\boldsymbol{x}_t}} \exp\left(\frac{1}{\alpha} \sum_{t=0}^{T-1} Q^*(\boldsymbol{x}_t, \boldsymbol{u}_t)\right) \tag{17b}$$

$$= \frac{1}{\prod_{t=0}^{T-1} Z_{\boldsymbol{x}_t}} \exp\left(\frac{1}{\alpha} \sum_{t=0}^{T-1} \left(r(\boldsymbol{x}_t, \boldsymbol{u}_t) + \alpha \log \int_{\mathcal{U}} \exp\left(\frac{1}{\alpha} Q^*(\boldsymbol{x}_{t+1}, \boldsymbol{u}_{t+1})\right) d\boldsymbol{u}\right)\right) \tag{17c}$$

$$= \frac{1}{\prod_{t=0}^{T-1} Z_{\boldsymbol{x}_t}} \exp\left(\frac{1}{\alpha} \sum_{t=0}^{T-1} r(\boldsymbol{x}_t, \boldsymbol{u}_t) + \log\left(\prod_{t=1}^{T-1} Z_{\boldsymbol{x}_t}\right) + \log Z_{\boldsymbol{x}_T}\right) \tag{17d}$$

$$= \frac{\prod_{t=1}^{T-1} Z_{\boldsymbol{x}_t}}{\prod_{t=0}^{T-1} Z_{\boldsymbol{x}_t}} \exp\left(\frac{1}{\alpha} \left(\sum_{t=0}^{T-1} r(\boldsymbol{x}_t, \boldsymbol{u}_t) + \alpha \log Z_{\boldsymbol{x}_T}\right)\right) \tag{17e}$$

$$= \frac{1}{Z_{\boldsymbol{x}_0}} \exp\left(\frac{1}{\alpha}\left(\sum_{t=0}^{T-1} r(\boldsymbol{x}_t, \boldsymbol{u}_t) + V^*(\boldsymbol{x}_T)\right)\right), \tag{17f}$$

where Eq. equation 17b results from substituting Eq. equation 14 and Eq. equation 17c results from substituting Eq. equation 15.

In Eq. equation 17f, $V^*(\boldsymbol{x}_T) = \alpha \log Z_{\boldsymbol{x}_T}$ is the soft value function defined in (Haarnoja et al., 2017) at the terminal step. Each $\boldsymbol{x}_t$ is defined recursively from $\boldsymbol{x}_0$ and $U$ through the dynamics model $F$ as $\boldsymbol{x}_{t+1} = F(\boldsymbol{x}_t, \boldsymbol{u}_t), t = 0, \cdots, T-1$. QED

## B COMPLETE RESIDUAL-MPPI DEPLOYMENT PIPELINE

The complete Residual-MPPI deployment pipeline is shown in Algorithm 2

---
**Algorithm 2** Residual-MPPI Deployment Pipeline
---
**Require:** Prior policy $\pi$
1: Initialize a dataset $\mathcal{D} \leftarrow \emptyset$, dynamics $F_\theta$
2: **for** $t = 0, 1, \dots$ **do**          ▷ Dynamics Training
3:      $\boldsymbol{u}_t = \arg\max \pi(\boldsymbol{u}|\boldsymbol{x}_t) + \mathcal{E}$          ▷ *Exploration Noise*
4:      $\boldsymbol{x}_{t+1} \leftarrow \text{Environment}(\boldsymbol{x}_t, \boldsymbol{u}_t)$
5:      $\mathcal{D} \leftarrow \mathcal{D} \cup (\boldsymbol{x}_t, \boldsymbol{u}_t, \boldsymbol{x}_{t+1})$          ▷ *Multi-step Error*
6:      $\theta \leftarrow \arg\min_\theta \mathbb{E}_{\mathcal{D}} \left[\sum_{t=0}^{T} \gamma^t (\boldsymbol{x}_{t+1} - F_\theta(\hat{\boldsymbol{x}}_t, \boldsymbol{u}_t))^2\right]$
7: **end for**
8: **for** $t = 0, 1, \dots$ **do**          ▷ Zero-shot Residual-MPPI
9:      $\boldsymbol{u}_t = \text{Residual-MPPI}(\boldsymbol{x}_t)$
10:      $\boldsymbol{x}_{t+1} \leftarrow \text{Environment}(\boldsymbol{x}_t, \boldsymbol{u}_t)$
11: **end for**
12: **for** $t = 0, 1, \dots$ **do**          ▷ Few-shot Residual-MPPI
13:      $\boldsymbol{u}_t = \text{Residual-MPPI}(\boldsymbol{x}_t)$
14:      $\boldsymbol{x}_{t+1} \leftarrow \text{Environment}(\boldsymbol{x}_t, \boldsymbol{u}_t)$
15:      $\mathcal{D} \leftarrow \mathcal{D} \cup (\boldsymbol{x}_t, \boldsymbol{u}_t, \boldsymbol{x}_{t+1})$          ▷ *Fine-tune with Online Data*
16:      $\theta \leftarrow \arg\min_\theta \mathbb{E}_{\mathcal{D}} \left[\sum_{t=0}^{T} \gamma^t (\boldsymbol{x}_{t+1} - F_\theta(\hat{\boldsymbol{x}}_t, \boldsymbol{u}_t))^2\right]$
17: **end for**
---

## C IMPLEMENTATION DETAILS IN MUJOCO

All the experiments were conducted on Ubuntu 22.04 with Intel Core i9-9920X CPU @ 3.50GHz × 24, NVIDIA GeForce RTX 2080 Ti, and 125 GB RAM.

### C.1 MuJoCo Environment Configuration

In this section, we introduce the detailed configurations of the selected environments, including the basic tasks, add-on tasks, and the corresponding rewards. The action and observation space of all the environments follow the default settings in `gym[mujoco]-v3`.

**Half Cheetah.** In the `HalfCheetah` environment, the basic goal is to apply torque on the joints to make the cheetah run forward (right) as fast as possible. The state and action space has 17 and 6 dimensions, and the action represents the torques applied between links.

The basic reward consists of two parts:

$$\text{Forward Reward} : r_{\text{forward}}(\boldsymbol{x}_t, \boldsymbol{u}_t) = \frac{\Delta x}{\Delta t}$$
$$\text{Control Cost} : r_{\text{control}}(\boldsymbol{x}_t, \boldsymbol{u}_t) = -0.1 \times ||\boldsymbol{u}_t||_2^2 \tag{18}$$

During policy customization, we demand an additional task that requires the cheetah to limit the angle of its hind leg. This customization goal is formulated as an add-on reward function defined as:

$$r_R(\boldsymbol{x}_t, \boldsymbol{u}_t) = -10 \times |\theta_{\text{hind leg}}| \tag{19}$$

**Hopper.** In the `Hopper` environment, the basic goal is to make the hopper move in the forward direction by applying torques on the three hinges connecting the four body parts. The state and action space has 11 and 3 dimensions, and the action represents the torques applied between links.

The basic reward consists of three parts:

$$\text{Alive Reward} : r_{\text{alive}} = 1$$
$$\text{Forward Reward} : r_{\text{forward}}(\boldsymbol{x}_t, \boldsymbol{u}_t) = \frac{\Delta x}{\Delta t} \tag{20}$$
$$\text{Control Cost} : r_{\text{control}}(\boldsymbol{x}_t, \boldsymbol{u}_t) = -0.001 \times ||\boldsymbol{u}_t||_2^2$$

The episode will terminate if the z-coordinate of the hopper is lower than 0.7, or the angle of the top is no longer contained in the closed interval $[-0.2, 0.2]$, or an element of the rest state is no longer contained in the closed interval $[-100, 100]$.

During policy customization, we demand an additional task that requires the hopper to jump higher along the z-axis. This customization goal is formulated as an add-on reward function defined as:

$$r_R(\boldsymbol{x}_t, \boldsymbol{u}_t) = 10 \times (z - 1) \tag{21}$$

**Swimmer.** In the `Swimmer` environment, the basic goal is to move as fast as possible towards the right by applying torque on the rotors. The state and action space has 8 and 2 dimensions, and the action represents the torques applied between links.

The basic reward consists of two parts:

$$\text{Forward Reward} : r_{\text{forward}}(\boldsymbol{x}_t, \boldsymbol{u}_t) = \frac{\Delta x}{\Delta t}$$
$$\text{Control Cost} : r_{\text{control}}(\boldsymbol{x}_t, \boldsymbol{u}_t) = -0.0001 \times ||\boldsymbol{u}_t||_2^2 \tag{22}$$

During policy customization, we demand an additional task that requires the agent to limit the angle of its first rotor. This customization goal is formulated as an add-on reward function defined as:

$$r_R(\boldsymbol{x}_t, \boldsymbol{u}_t) = -1 \times |\theta_{\text{first rotor}}| \tag{23}$$

**Ant.** In the `Ant` environment, the basic goal is to coordinate the four legs to move in the forward (right) direction by applying torques on the eight hinges connecting the two links of each leg and the torso (nine parts and eight hinges). The state and action space has 27 and 8 dimensions, and the action represents the torques applied at the hinge joints.

The basic reward consists of three parts:

$$\text{Alive Reward} : r_{\text{alive}} = 1$$
$$\text{Forward Reward} : r_{\text{forward}}(\boldsymbol{x}_t, \boldsymbol{u}_t) = \frac{\Delta x}{\Delta t} \tag{24}$$
$$\text{Control Cost} : r_{\text{control}}(\boldsymbol{x}_t, \boldsymbol{u}_t) = -0.5 \times ||\boldsymbol{u}_t||_2^2$$

Table 3: RL Prior Policy Training Hyperparameters

| Hyperparameter | Value |
|---|---|
| Hidden Layers | $(256, 256)$ |
| Activation | $ReLu$ |
| $\gamma$ | 0.99 |
| Target Update Interval | 50 |
| Learning Rate | $3e - 4$ |
| Gradient Step | 1 |
| Training Frequency | 1 |
| Batch Size | 256 |
| Optimizer | $Adam$ |
| Total Samples | 6400000 |
| Capacity | 1000000 |
| Sampling | $Uniform$ |

Table 4: MuJoCo Offline Dynamics Training Hyperparameters

| Hyperparameter | Value |
|---|---|
| Hidden Layers | $(256, 256, 256, 256)$ |
| Activation | $Mish$ |
| learning rate | $1e - 5$ |
| Training Frequency | 10 |
| Optimizer | $Adam$ |
| Batch Size | 256 |
| Horizon | 8 |
| $\gamma$ | 0.9 |
| Total Samples | 200000 |
| Capacity | 50000 |
| Sampling | $Uniform$ |

The episode will terminate if the z-coordinate of the torso is not in the closed interval $[0.2, 1]$. During experiments, we set the upper bound to $\inf$ for both prior policy training and planning experiments as it benefits both performances.

During policy customization, we demand an additional task that requires the ant to move along the y-axis. This customization goal is formulated as an add-on reward function defined as:

$$r_R(\boldsymbol{x}_t, \boldsymbol{u}_t) = \frac{\Delta y}{\Delta t} \tag{25}$$

## C.2 RL Prior Policy Training

The prior policies were constructed using Soft Actor-Critic (SAC) with the StableBaseline3 (Raffin et al., 2021) implementation. The training was conducted in parallel across 32 environments. The hyperparameters used for RL prior policies training are shown in Table 3. Since this parameter setting performed poorly in the Swimmer task, we used the official benchmark checkpoint of Swimmer from StableBaseline3. Note that the prior policies do not necessarily need to be synthesized using RL. We report the experiment results with GAIL prior policies in Appendix F.1.

## C.3 Offline Dynamics Training

The hyperparameters used for offline dynamics training are shown in Table 4. The exploration noise we utilized comes from the guassian distribution of the prior policy.

## C.4 Planning Hyperparameters

The hyperparameters used for online planning are shown in Table 5. At each step, the planners computes the result based on the current observation. Only the first action of the computed action sequence is sent to the system.

Table 5: Planning Hyperparameter in MuJoCo Tasks

| Hyperparameter | Half Cheetah | Ant | Swimmer | Hopper |
|---|---|---|---|---|
| Horizon | 2 | 5 | 5 | 8 |
| Samples | 10000 | 5000 | 5000 | 10000 |
| Noise std. | 0.017 | 0.02 | 0.005 | 0.005 |
| $\omega'$ | $1e-7$ | $1e-2$ | $1e-4$ | $2e-7$ |
| $\gamma$ | 0.9 | 0.9 | 0.9 | 0.9 |
| $\lambda$ | $5e-5$ | $5e-3$ | $1e-4$ | $1e-5$ |

## D  IMPLEMENTATION DETAILS IN GTS

All the GTS experiments were conducted on PlayStation 5 (PS5) and Ubuntu 20.04 with 12th Gen Intel Core i9-12900F $\times$ 24, NVIDIA GeForce RTX 3090, and 126 GB RAM. GTS ran independently on PS5 with a fixed frequency of 60Hz. The communication protocol returned the observation and received the control input with a frequency of 10Hz.

### D.1  GTS ENVIRONMENT CONFIGURATION

The action and observation spaces follow the configuration of GT Sophy 1.0 (Wurman et al., 2022). The reward used for GT Sophy 1.0 training was a handcrafted linear combination of reward components computed on the transition between the previous and current states, which included course progress, off-course penalty, wall penalty, tire-slip penalty, passing bonus, any-collision penalty, rear-end penalty, and unsporting-collision penalty (Wurman et al., 2022). During policy customization, we demand an additional task that requires the GT Sophy 1.0 to drive on course. This customization goal is formulated as an add-on reward function defined as:

$$r_R(\boldsymbol{x}_t, \boldsymbol{u}_t) = -1000000 \times ReLu(d_{\text{center}}^2 - d_{\text{map}}^2), \quad (26)$$

where $d_{\text{center}}$ is the distance from the vehicle to the course center and $d_{\text{map}}$ is the width of the course given the car's position.

### D.2  DYNAMICS DESIGN

We employed three main techniques to help us get an accurate dynamics model.

**Historical Observations** To address the partially observable Markov decision processes (POMDP) nature of the problem, we included historical observations in the input to help the model capture the implicit information.

**Splitting State Space** We divided the state space into two parts: the *dynamic states* and *map information*. We adopted a neural network with an MLP architecture to predict the *dynamic states*. In each step, we utilized the trained model to predict the *dynamic states* and leveraged the known map to calculate the *map information* based on the *dynamic states*.

**Physical Prior** Some physical states in the *dynamic states*, such as wheel load and slip angle, are intrinsically difficult to predict due to the limited observation. To reduce the variance brought by these difficult states, two neural networks were utilized to predict them and other states independently.

Table 6 shows the hyperparameters used for training these two MLPs to predict *dynamic states*.

### D.3  PLANNING HYPERPARAMETERS

To stabilize the planning process, we further utilized another hyperparameter $topratio$ to select limited action sequence candidates with top-tier accumulated reward. The hyperparameters used for online planning are shown in Table 7. At each step, the planners computed the result based on the current observation. Only the first action of the computed action sequence was sent to the system.

### D.4  RESIDUAL-SAC TRAINING

In addition to the Residual-SAC algorithm, we adopted the idea of residual policy learning (Silver et al., 2018) to learn a policy that corrects the action of GT Sophy 1.0 by outputting an additive

Table 6: GTS Offline Dynamics Training Hyperparameters

| Hyperparameter | Value |
| --- | --- |
| History Length | 8 |
| Hidden Layers | $(2048, 2048, 2048)$ |
| Activation | $Mish$ |
| Learning Rate | $1e-5$ |
| Training Steps | 200000 |
| Training Frequency | 5 |
| Batch Size | 256 |
| Horizon | 5 |
| $\gamma$ | 1 |
| Optimizer | $Adam$ |
| Capacity | 2000000 |
| Sampling | $Uniform$ |

Table 7: Planning Hyperparameter in GTS

| Hyperparameter | Horizon | Samples | Noise std. | Top Ratio | $\omega'$ | $\gamma$ | $\lambda$ |
| --- | --- | --- | --- | --- | --- | --- | --- |
| Value | 15 | 500 | 0.035 | 0.048 | 3 | 0.8 | 0.5 |

action, in which the initial policy was set as GT Sophy 1.0. As proposed in the same paper, the weights of the last layer in the actor network were initiated to be zero. The randomly initialized critic was trained alone with a fixed actor first. And then, both networks were trained together. The pipeline was developed upon the official implementation of RQL (Li et al., 2024). The training hyperparameters are shown in Table 8.

Table 8: Residual-SAC Training Hyperparameters

| Hyperparameter | Value |
| --- | --- |
| Hidden Layers | $(2048, 2048, 2048)$ |
| Activation | $ReLu$ |
| Learning Rate | $1e-4$ |
| Target Update Interval | 50 |
| Gradient Step | 1 |
| Training Frequency | 1 |
| Batch Size | 256 |
| Optimizer | $Adam$ |
| $\gamma$ | 0.9 |
| Total Samples | 10000000 |
| Capacity | 1000000 |
| Sampling | $Uniform$ |

# E  VISUALIZATION

## E.1  MUJOCO EXPERIMENTS

We visualize some representative running examples from the MuJoCo environment in Figure 4. As shown in the plot, the customized policies achieved significant improvements over the prior policies in meeting the objectives of the add-on tasks.

## E.2  GTS EXPERIMENTS

We provide four typical complete trajectories of all policies in Figure 5. Also, we visualize the sim-to-sim experiment results in Figure 6. The visualization clearly demonstrates the effectiveness of the proposed method by greatly reducing the off-course steps of a champion-level driving policy.

We visualize the racing line selected by each policy at four typical corners to illustrate the differences among policies. As shown in Figure 7, GT Sophy 1.0 exhibited aggressive racing lines that tend to be off course. Both Zero-shot MPPI and Few-shot MPPI were able to customize the behavior to drive

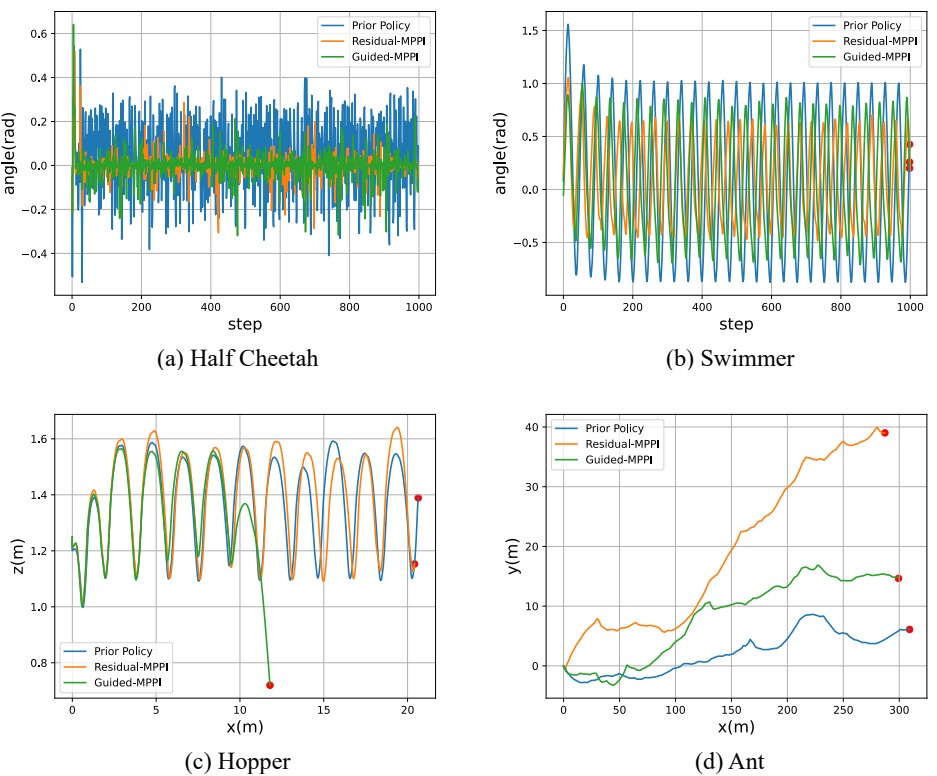

Figure 4: (a) The angle of the Half Cheetah's hind leg vs. the environmental steps. (b) The angle of the Swimmer's first rotor vs. the environmental steps. (c) The trajectory of the Hopper robot on the $x$ and $z$ axis.(d) The trajectory of the Ant robot on the $x$ and $y$ axis.

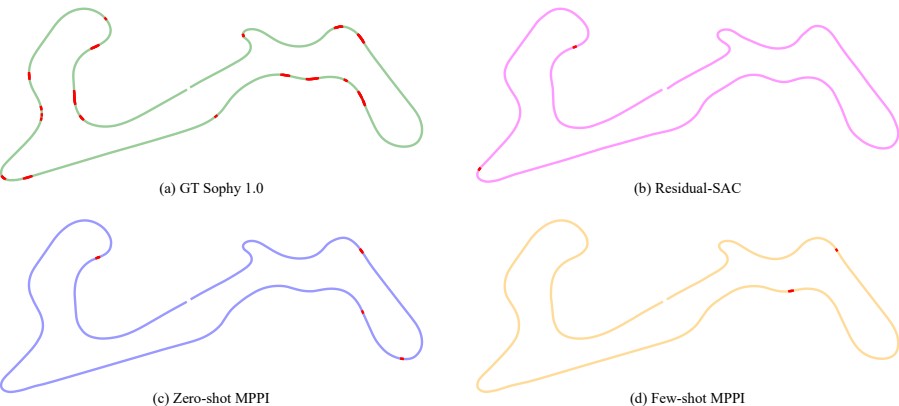

Figure 5: Typical complete trajectories of all policies, where the red parts indicate off-course behaviours. (a) The trajectory of GT Sophy 1.0. It finishes the lap in 117.762s with 93 steps off course. (b) The trajectory of Residual-SAC. It finishes the lap in 131.078s with 2 steps off course. (c) The trajectory of Zero-shot MPPI. It finishes the lap in 123.551s with 10 steps off course. (d) The trajectory of Few-shot MPPI. It finishes the lap in 122.919s with 4 steps off course.

more on the course, while the Few-shot MPPI chose a better racing line. In contrast, the Residual-SAC yields to an overly conservative behavior and keeps driving in the middle of the course.

Meanwhile, we visualize the difference between the actions selected by GT Sophy 1.0 and Residual-MPPI at each time step in Figure 8. When driving at the corner, our method exerted notable influ-

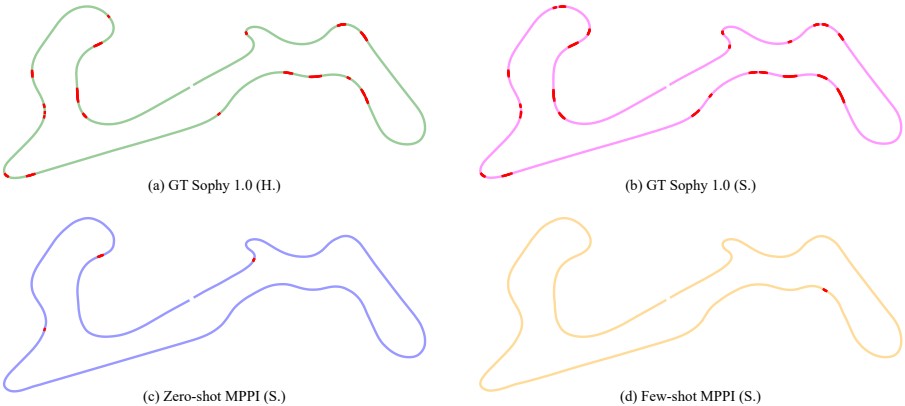

(a) GT Sophy 1.0 (H.)      (b) GT Sophy 1.0 (S.)

(c) Zero-shot MPPI (S.)      (d) Few-shot MPPI (S.)

Figure 6: Typical complete trajectories of all policies in Sim2Sim experiments, where the red parts indicate off-course behaviours. (a) The trajectory of GT Sophy 1.0 with Race-Hard. It finishes the lap in 117.762s with 93 steps off course. (b) The trajectory of Residual-SAC with Race-Soft. It finishes the lap in 116.674s with 133 steps off course. (c) The trajectory of Zero-shot MPPI with Race-Soft. It finishes the lap in 123.550s with 9 steps off course. (d) The trajectory of Few-shot MPPI with Race-Soft. It finishes the lap in 122.379s with 2 steps off course.

ences on the prior policy, as shown in Figure 8 (a). When driving at the straight, our method exerted minimal influences on the prior policy since most forward-predicted trajectories remain on course, as shown in Figure 8 (b).

## F ADDITIONAL EXPERIMENTAL RESULTS

### F.1 RESIDUAL-MPPI WITH IL PRIOR

Residual-MPPI is applicable to any prior policies trained based on the maximum-entropy principle, not limited to RL methods. In this section, we conduct additional experiments upon the same MuJoCo environments with IL prior policies. The IL prior policies are obtained through Generative Adversarial Imitation Learning (GAIL) with expert data generated by the RL prior in the previous experiments. The hyperparameters used for training the IL prior policies are shown in Table 9, while the hyperparameters used for the learners are the same as the corresponding RL experts.

Table 9: Hyperparameters of GAIL Imitated Polices

| Hyperparameter | Half Cheetah | Ant | Swimmer | Hopper |
|---|---|---|---|---|
| expert_min_episodes | 1000 | 100 | 1000 | 2000 |
| demo_batch_size | 512 | 25000 | 1024 | 1024 |
| gen_replay_buffer_capacity | 8192 | 50000 | 2048 | 2048 |
| n_disc_updates_per_round | 4 | 4 | 4 | 4 |

The results are summarized in Table 10. Similar to the experiment results with the RL priors, the customized policies achieved significant improvements over the prior policies in meeting the objectives of the add-on tasks, which demonstrates the applicability of Residual-MPPI with IL priors.

### F.2 ABLATION IN MUJOCO

We conducted ablation studies for Guided-MPPI and Residual-MPPI in MuJoCo with 100 episodes for each selected setting, whose evaluation logs are available in the supplementary materials.

**Horizon.** Regarding the question of why Guided-MPPI baseline underperforms the proposed approach, as we explained in the main text, while Guided-MPPI improves the estimation of optimal actions through more efficient sampling, it remains hindered by its ability to account for the long-term impacts of actions within finite planning horizons. In contrast, Residual-MPPI implicitly inherits

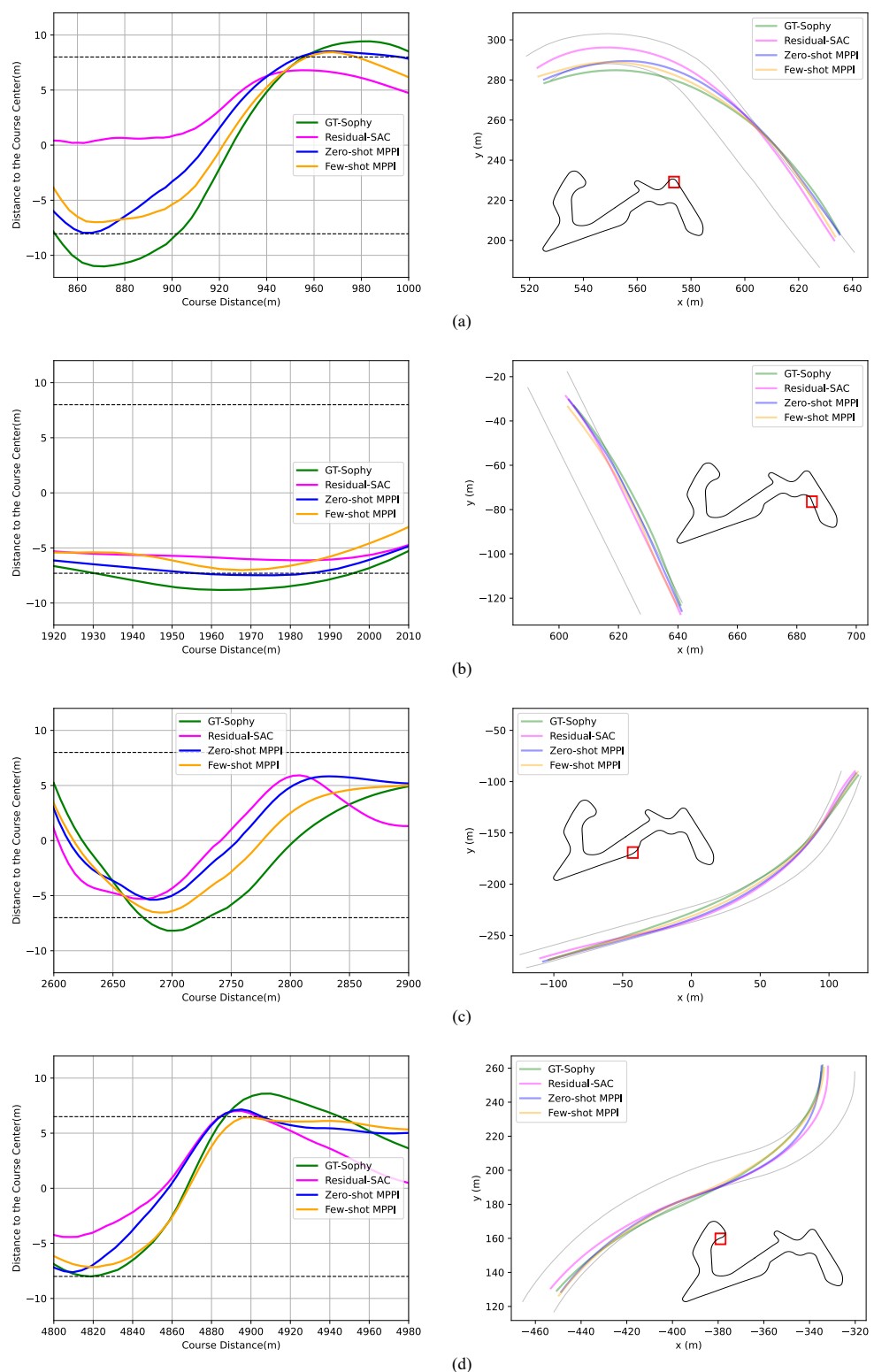

Figure 7: Route Selection Visualization at Different Cases. Different colors denote the different policies. In all cases, Residual-MPPI are able to customize the behavior to drive more on course. In (a), (b) and (d), due to inaccurate dynamics, Zero-shot MPPI exhibits off-course behavior. In (c), although Few-shot MPPI and Zero-shot MPPI both drive on course, Few-shot MPPI tends to select a better route closer to the boundary with more accurate dynamics.

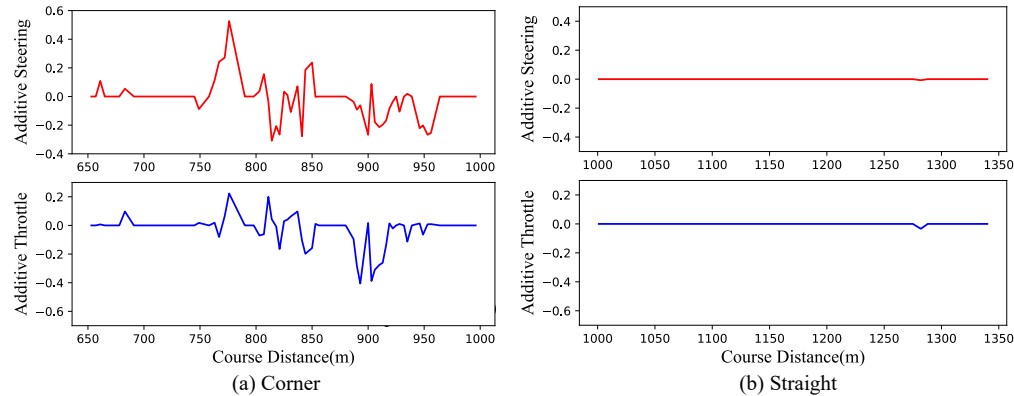

Figure 8: Additive Action of Residual-MPPI at Different Case. The additive throttle and steering are all linear normalized to $-1$ to $1$.

the information of the basic reward through the prior policy log-likelihood. And the prior policy log-likelihood is informed by the value functions optimized over an infinite horizon or demonstrations. Therefore, in our hypothesis, a longer horizon will benefit more on Guided-MPPI compared to Residual-MPPI. As shown in Figure 9, the performance of each algorithm matches our hypothesis. It is worth noting that Guided-MPPI still underperforms Residual-MPPI in all cases except in HalfCheetah. Meanwhile, in some environments, an overlong horizon decreases the performance. These results reflect the limitation of the planning-based methods with inaccurate dynamics.

**Samples.** The number of samples would have a lower impact as it primarily controls the sufficiency of optimization. When the number exceeds the threshold required for the optimization, further performance improvement can hardly be observed. From the results shown in Figure 10, we can see that the chosen number of samples is already sufficient for each optimization. However, the performance of Guided-MPPI in Hopper and Ant reduces with more samples, which is due to the error from the terminal value estimation. In this situation, Residual-MPPI demonstrates strong stability.

**Temperature.** The temperature controls the algorithm's tendency to average various actions or only focus on the top-tier actions. As shown in the Figure 11: a higher temperature can potentially lead to instability, as observed in the Hopper and Swimmer tasks; however, it can also allow more actions to be adequately considered, improving performance, as observed in the Ant task.

**Noise Std.** The Noise Std. parameter controls the magnitude of action noise, affecting the algorithm's exploration performance. As shown in Figure 12: a small noise would hinder the policy from exploring better actions, as observed in the Swimmer task; however, it can also increase stability, leading to higher overall performance, as observed in the Hopper task.

**Omega.** Theoretically, $\omega'$ represents the trade-off between the basic task and the add-on task. As shown in Figure 13: in complex tasks that the add-on reward is orthogonal to the basic reward (e.g., Ant), a larger $\omega'$ leads to behaviors close to the prior policy and higher basic reward; and a smaller $\omega'$ leads to behaviors close to the Greedy-MPPI and higher add-on reward but lower total reward.

### F.3 ABLATION IN GTS

We also conducted the corresponding ablation studies for Few-shot MPPI in GTS with 10 laps for each selected setting. Due to the strict experimental requirement of a 10Hz control frequency, we found the maximum number of samples that satisfies this requirement for each chosen horizon and conducted experiments. The results shown in Figure 14, Figure 15, and Figure 16 match our hypothesis.

In GTS, we also conduct an ablation on $\omega'$ parameter. Theorectically, $\omega'$ represents the trade-off between the basic task and the add-on task, where larger $\omega'$ should lead to a behavior closer to the prior policy with smaller lap time and larger off-course steps. On the other hand, oversmall $\omega'$ would construct a biased MDP, leading to suboptimal performace both on laptime and off-course steps. The corresponding ablation results in GTS are shown in Figure 17, which also follow our analysis.

Table 10: Experimental Results of Zero-shot Residual-MPPI with IL Prior in MuJoCo

| Env. | Policy | Full Task | Basic Task | Add-on Task | |
|---|---|---|---|---|---|
| | | Total Reward | Basic Reward | $|\bar{\theta}|$ | Add-on Reward |
| Half Cheetah | Prior Policy | $-971.3 \pm 689.1$ | $2288.9 \pm 725.5$ | $0.33 \pm 0.01$ | $-3260.3 \pm 93.5$ |
| | Greedy-MPPI | $\mathbf{-408.5 \pm 247.4}$ | $2397.9 \pm 285.3$ | $\mathbf{0.28 \pm 0.01}$ | $\mathbf{-2806.4 \pm 141.3}$ |
| | Full-MPPI | $-3569.7 \pm 353.9$ | $-1164.4 \pm 152.0$ | $0.24 \pm 0.03$ | $-2415.2 \pm 341.3$ |
| | Guided-MPPI | $-620.3 \pm 217.6$ | $2236.3 \pm 219.2$ | $0.29 \pm 0.01$ | $-2856.6 \pm 93.6$ |
| | Valued-MPPI | $-643.7 \pm 224.3$ | $\mathbf{2231.7 \pm 202.9}$ | $0.29 \pm 0.01$ | $-2875.5 \pm 79.9$ |
| | Residual-MPPI | $\mathbf{-415.2 \pm 265.5}$ | $2383.7 \pm 303.6$ | $\mathbf{0.28 \pm 0.01}$ | $\mathbf{-2798.9 \pm 145.1}$ |
| | Residual-SAC (200K) | $-356.0 \pm 77.0$ | $703.6 \pm 71.5$ | $0.11 \pm 0.00$ | $-1059.7 \pm 30.1$ |
| | Residual-SAC (4M) | $2111.5 \pm 79.1$ | $2382.3 \pm 78.0$ | $0.03 \pm 0.00$ | $-270.8 \pm 7.3$ |

| Env | Policy | Total Reward | Basic Reward | $|\bar{\theta}|$ | Add-on Reward |
|---|---|---|---|---|---|
| Swimmer | Prior Policy | $-344.5 \pm 3.3$ | $328.2 \pm 1.5$ | $0.67 \pm 0.00$ | $-672.7 \pm 1.9$ |
| | Greedy-MPPI | $\mathbf{-35.1 \pm 7.1}$ | $232.7 \pm 3.8$ | $\mathbf{0.27 \pm 0.01}$ | $\mathbf{-267.8 \pm 7.3}$ |
| | Full-MPPI | $-1685.3 \pm 108.1$ | $13.5 \pm 7.5$ | $1.70 \pm 0.11$ | $-1698.8 \pm 107.4$ |
| | Guided-MPPI | $-122.6 \pm 6.9$ | $222.1 \pm 4.8$ | $0.34 \pm 0.01$ | $-344.7 \pm 7.6$ |
| | Valued-MPPI | $-157.4 \pm 6.5$ | $\mathbf{243.8 \pm 5.0}$ | $0.40 \pm 0.01$ | $-401.3 \pm 8.1$ |
| | Residual-MPPI | $\mathbf{-35.1 \pm 6.7}$ | $232.8 \pm 4.2$ | $\mathbf{0.27 \pm 0.01}$ | $\mathbf{-267.9 \pm 7.2}$ |
| | Residual-SAC (200K) | $-168.1 \pm 113.8$ | $-0.4 \pm 18.1$ | $0.17 \pm 0.11$ | $-167.7 \pm 114.7$ |
| | Residual-SAC (4M) | $-9.3 \pm 15.4$ | $-0.1 \pm 14.3$ | $0.01 \pm 0.01$ | $-9.2 \pm 7.3$ |

| Env. | Policy | Total Reward | Basic Reward | $\bar{z}$ | Add-on Reward |
|---|---|---|---|---|---|
| Hopper | Prior Policy | $6790.9 \pm 40.2$ | $3439.8 \pm 12.7$ | $1.34 \pm 0.00$ | $3351.1 \pm 29.4$ |
| | Greedy-MPPI | $\mathbf{6881.8 \pm 180.6}$ | $3423.9 \pm 73.5$ | $\mathbf{1.35 \pm 0.00}$ | $\mathbf{3457.9 \pm 109.5}$ |
| | Full-MPPI | $20.7 \pm 3.2$ | $3.6 \pm 0.7$ | $1.24 \pm 0.00$ | $17.1 \pm 2.6$ |
| | Guided-MPPI | $6793.8 \pm 301.1$ | $3422.2 \pm 122.1$ | $1.34 \pm 0.01$ | $3370.8 \pm 183.1$ |
| | Valued-MPPI | $6832.2 \pm 179.6$ | $\mathbf{3434.7 \pm 73.3}$ | $1.34 \pm 0.00$ | $3397.4 \pm 108.3$ |
| | Residual-MPPI | $\mathbf{6902.8 \pm 41.0}$ | $3430.8 \pm 12.8$ | $\mathbf{1.35 \pm 0.00}$ | $\mathbf{3472.0 \pm 30.7}$ |
| | Residual-SAC (200K) | $5993.2 \pm 1327.7$ | $3019.0 \pm 627.6$ | $1.35 \pm 0.02$ | $2974.1 \pm 709.0$ |
| | Residual-SAC (4M) | $7077.2 \pm 514.9$ | $3445.9 \pm 229.2$ | $1.37 \pm 0.00$ | $3631.3 \pm 287.0$ |

| Env | Policy | Total Reward | Basic Reward | $\bar{v}_y$ | Add-on Reward |
|---|---|---|---|---|---|
| Ant | Prior Policy | $4169.1 \pm 1468.3$ | $4782.9 \pm 1639.1$ | $-0.61 \pm 0.22$ | $-613.7 \pm 216.4$ |
| | Greedy-MPPI | $\mathbf{4518.3 \pm 1755.3}$ | $4018.8 \pm 1597.1$ | $\mathbf{0.50 \pm 0.18}$ | $\mathbf{499.4 \pm 181.6}$ |
| | Full-MPPI | $-2780.7 \pm 153.7$ | $-2774.4 \pm 110.1$ | $-0.01 \pm 0.11$ | $-6.2 \pm 108.5$ |
| | Guided-MPPI | $3041.5 \pm 2111.2$ | $3098.6 \pm 2142.3$ | $-0.06 \pm 0.14$ | $-57.1 \pm 139.4$ |
| | Valued-MPPI | $4321.5 \pm 1848.7$ | $\mathbf{4727.7 \pm 1995.1}$ | $-0.41 \pm 0.22$ | $-406.2 \pm 222.6$ |
| | Residual-MPPI | $\mathbf{4877.6 \pm 1648.4}$ | $4565.9 \pm 1560.6$ | $\mathbf{0.31 \pm 0.14}$ | $\mathbf{311.6 \pm 137.2}$ |
| | Residual-SAC (200K) | $-180.3 \pm 20.4$ | $-181.5 \pm 20.4$ | $0.00 \pm 0.00$ | $1.1 \pm 1.1$ |
| | Residual-SAC (4M) | $6016.9 \pm 1001.8$ | $3836.1 \pm 705.3$ | $2.18 \pm 0.30$ | $2180.8 \pm 303.7$ |

The evaluation results are in the form of $\mathrm{mean} \pm \mathrm{std}$ over the 500 running episodes. The total reward is calculated on full task, whose reward is $\omega r + r_R$.

## F.4 Ablation of Dynimic Model

To demonstrate the effectiveness of the proposed dynamics training design, we conducted the corresponding ablation study. The results in Table 11 show that planning based on dynamics trained with a multi-step (5-step) error is more effective in both the laptime and off-course steps metrics. Moreover, using online data to fine-tune the dynamics model also yields significant performance improvements in the one-step error approach. Regarding the exploration noise, the effect of this technique in GTS is not particularly significant.

## F.5 Sim-to-sim Experiments

To motivate future sim-to-real deployments, we designed a preliminary sim-to-sim experiment by replacing the test vehicle's tires from Race-Hard (H.) to Race-Soft (S.) to validate the proposed algorithm's robustness under suboptimal dynamics and prior policy. Since Residual-MPPI is a model-based receding-horizon control algorithm. The dynamic replanning mechanism and dynamics model

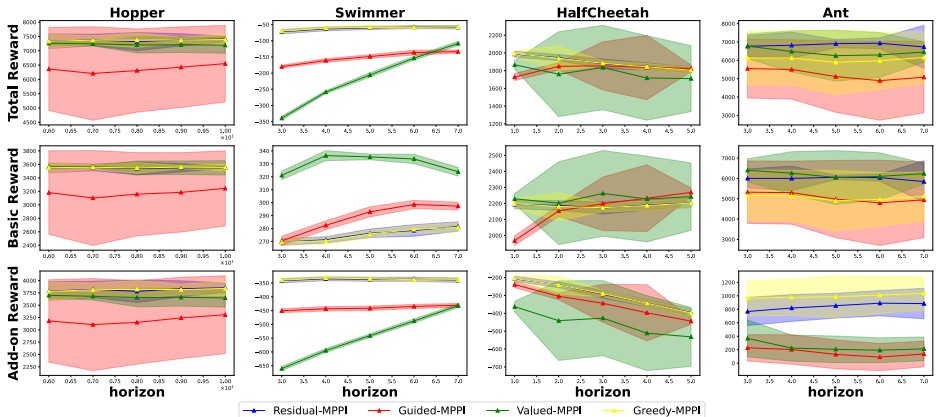

Figure 9: Ablation Study of Horizon in MuJoCo

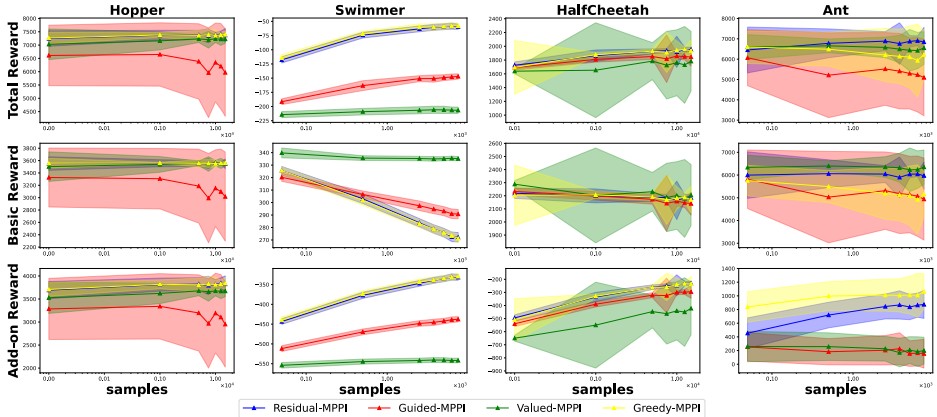

Figure 10: Ablation Study of Samples in MuJoCo

adaptation could potentially enable robust and adaptive domain transfer under tire dynamics discrepancy. The experimental setup and parameter selection are consistent with those in Sec. 5.2.

In sim-to-sim experiments, the domain gap brought by the tire leads to massively increased off-course steps in prior policy, which may lead to severe accidents in sim-to-real applications. In contrast, Residual-MPPI could still drive the car safely on course with minor speed loss, despite the suboptimality in prior policy and learned dynamics. Furthermore, as it shown in the Few-shot MPPI (S.) results, Residual-MPPI could serve as a safe starting point for data collection, policy finetuning, and possible future sim-to-real applications.

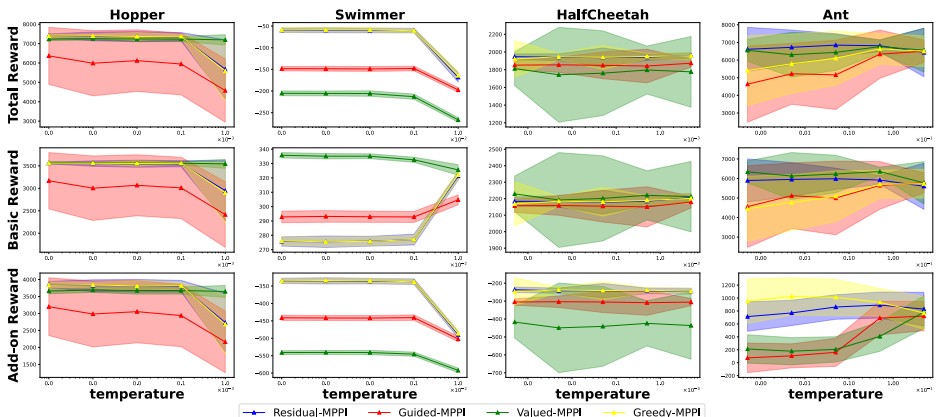

Figure 11: Ablation Study of Tempreture in MuJoCo

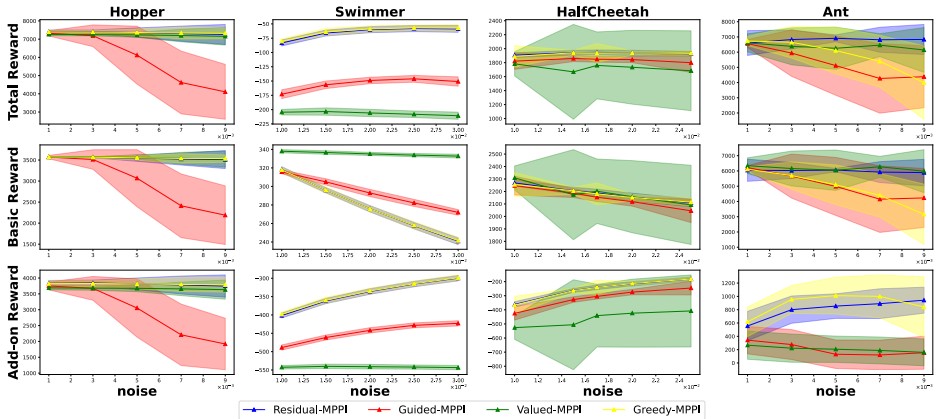

Figure 12: Ablation Study of Noise Std. in MuJoCo

### F.6 COMPARISON WITH PROMPT-DT

As discussed in the related work, Prompt-DT tries to solve the few-shot adaptation problem. Some readers may get interested in the differences between policy customization and few-shot adaptation. In this section, we provide a detailed comparison of Prompt-DT and Residual-MPPI both in theory and experiments.

Comparing to policy customization, few-shot adaptation seeks to adapt a policy solely to the new reward and requires full knowledge of the reward function. Moreover, few-shot adaptation scheme proposed in Prompt-DT relies on a DecisionTransformer-based prior policy trained on multi-task expert demonstration data via offline RL.

To make the comparison as fair as possible, we bring Prompt-DT to a setting closer to the policy customization. We select the ANT-dir environment and test Prompt-DT with the official implementation and its training parameters. Similar to the Ant environment in our original experiments, we define the basic reward as the progress along the x-axis, and the add-on reward as the progress along the y-axis.

First, we train a Prompt-DT model through multi-task offline RL as the prior policy. The multi-task demonstration data consists of expert trajectories walking along various directions (see *Prompt-DT*

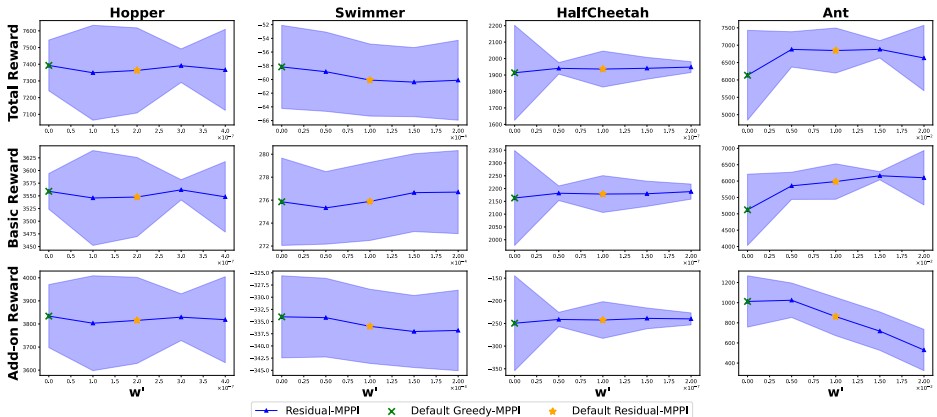

Figure 13: Ablation Study of $\omega'$ in MuJoCo

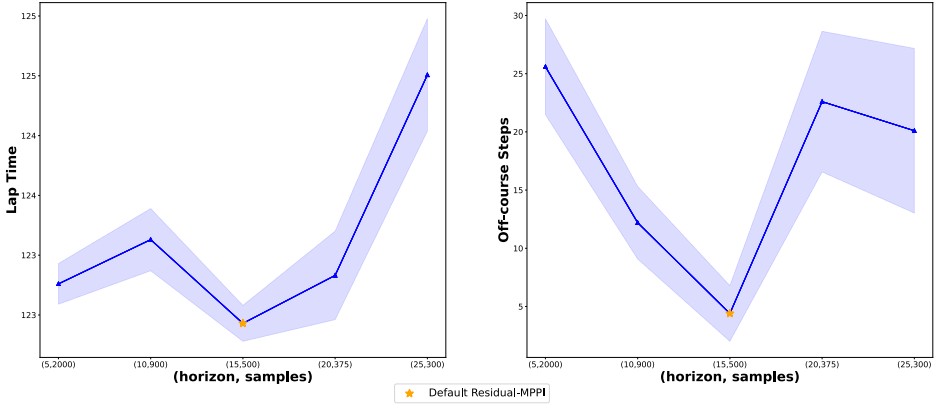

Figure 14: Ablation Study of Horizon and Samples in GTS

*Basic Task Demo* in Table 13 for the rewards of the demo trajectories provided for the basic task). Prompt-DT can be directed to solve the basic task, by conditioning on a trajectory prompt that is 1) sampled from the basic task demo, and 2) labeled with the basic reward function (see *Prompt-DT on Basic Task* in Table 13 for its performance).

We construct two types of prompts to adapt Prompt-DT to the full task defined by the sum of basic and add-on rewards:

- **Expert Prompt**: the trajectory prompt that is 1) collected with an *expert policy* trained on the full task, and 2) labeled with the *total reward* function. This prompt follows the original Prompt-DT setting. However, it has priviledged access to an expert policy and the total reward function, which are not given in policy customization.

- **Prior Prompt**: the trajectory prompt that is 1) sampled from the basic task demo, and 2) labeled with the add-on reward function. It is the modification suggested by the reviewer to bring Prompt-DT closer to our setting.

The performance of the customized policies is reported in Table 14. If provided with the prior prompt, Prompt-DT can hardly customize itself to the full task, with minor improvements on the add-on task. Moreover, even when provided with the expert prompt, Prompt-DT fails to strike a

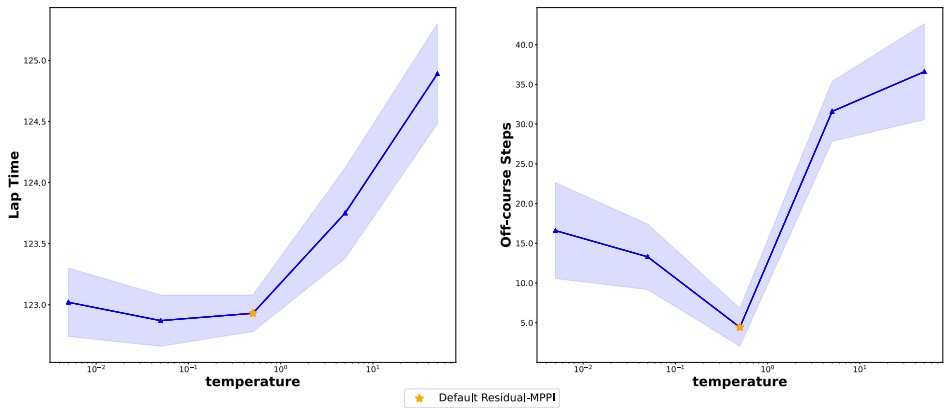

Figure 15: Ablation Study of Temperature in GTS

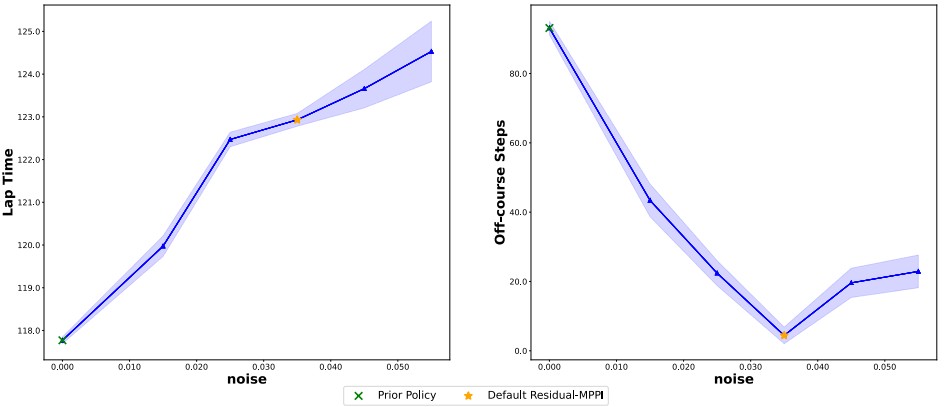

Figure 16: Ablation Study of Noise in GTS

good trade-off—-it sacrifices too much on the basic reward to get a higher add-on reward. The total reward is even lower than the case with the prior prompt. In contrast, Residua-MPPI customizes the prior policy to achieve a higher add-on reward with minor degradation on the basic task, leading to a much higher total reward.

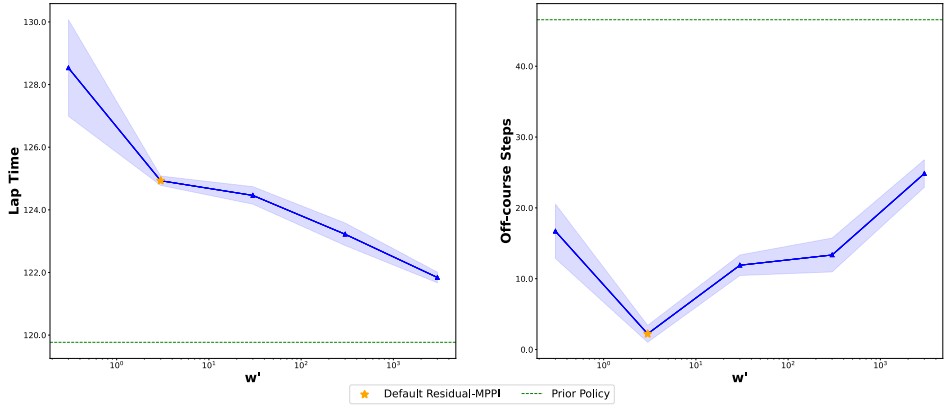

Figure 17: Ablation Study of $\omega'$ in GTS

Table 11: Dynamics Design Ablation Results in GTS

| Policy | Modules | | | Add-on Task | |
|---|---|---|---|---|---|
| | Exploration Noise | Multi-Step Error | Finetune | Lap Time | Off-course Steps |
| one-step Zero-shot MPPI | ✓ | ✗ | ✗ | $124.2 \pm 0.4$ | $13.2 \pm 5.3$ |
| one-step Few-shot MPPI | ✓ | ✗ | ✓ | $123.3 \pm 0.1$ | $7.1 \pm 3.8$ |
| No-exp Zero-shot MPPI | ✗ | ✓ | ✗ | $123.5 \pm 0.3$ | $10.0 \pm 3.7$ |
| No-exp Few-shot MPPI | ✗ | ✓ | ✓ | $123.0 \pm 0.2$ | $4.7 \pm 2.9$ |
| Zero-shot MPPI | ✓ | ✓ | ✗ | $123.3 \pm 0.2$ | $9.0 \pm 3.3$ |
| Few-shot MPPI | ✓ | ✓ | ✓ | $\mathbf{122.9 \pm 0.1}$ | $\mathbf{4.4 \pm 2.4}$ |

The evaluation results are in the form of $\mathrm{mean} \pm \mathrm{std}$ over 10 laps.

Table 12: Sim-to-Sim Experimental Results of Residual-MPPI in GTS

| Policy | GT Sophy 1.0 (H.) | GT Sophy 1.0 (S.) | Zero-shot MPPI (S.) | Few-shot MPPI (S.) |
|---|---|---|---|---|
| Lap Time | $117.7 \pm 0.1$ | $116.8 \pm 0.1$ | $123.5 \pm 0.2$ | $122.6 \pm 0.3$ |
| Off-course Steps | $93.1 \pm 2.0$ | $131.5 \pm 2.7$ | $8.3 \pm 3.6$ | $3.9 \pm 2.9$ |

The evaluation results are in the form of $\mathrm{mean} \pm \mathrm{std}$ over 10 laps.

Table 13: Evaluation of Prior Polices in Residual-MPPI and Prompt-DT

| Prior Policy | Total Reward | Basic Reward | Add-on Reward |
|---|---|---|---|
| Residual-MPPI Prior | $784.9 \pm 127.4$ | $799.7 \pm 108.5$ | -14.8 $\pm$ 58.5 |
| Prompt-DT Basic Task Demo | $779.5 \pm 109.0$ | $793.0 \pm 61.7$ | -13.5 $\pm$ 25.0 |
| Prompt-DT on Basic Task | $686.4 \pm 109.0$ | $691.8 \pm 101.1$ | -5.4 $\pm$ 45.5 |

The evaluation results are in the form of $\mathrm{mean} \pm \mathrm{std}$ over 500 episodes.

Table 14: Evaluation of Customized Polices in Residual-MPPI and Prompt-DT

| Customized Policy | Total Reward | Basic Reward | Add-on Reward |
|---|---|---|---|
| Prompt-DT (Prior Prompt) | $678.4 \pm 126.5$ | $677.0 \pm 114.3$ | $1.3 \pm 46.3$ |
| Prompt-DT (Expert Prompt) | $626.0 \pm 187.8$ | $605.6 \pm 184.8$ | $20.4 \pm 55.5$ |
| Residual-MPPI | $872.0 \pm 69.0$ | $774.8 \pm 41.8$ | $97.1 \pm 48.5$ |

The evaluation results are in the form of $\mathrm{mean} \pm \mathrm{std}$ over 500 episodes.

