# OpenReview forum: "Residual-MPPI: Online Policy Customization for Continuous Control"
_ICLR.cc/2025/Conference — ICLR 2025 Poster_

### Official Review · Reviewer_VRbb · 2024-10-18

**Soundness:** 3
**Presentation:** 2
**Contribution:** 3
**Rating:** 6
**Confidence:** 4

**Summary:**

This paper introduces an algorithm, Residual-MPPI, for customizing continuous control policies at execution time, without fine-tuning, in order to perform better at an auxiliary task while maintaining performance on the original task. Residual-MPPI does not require knowledge of the original task reward or value function; only the prior policy. However, the algorithm requires learning a dynamics model if one is not available, requiring additional online data to train and possibly iteratively improve as the policy changes (although apparently less data is required than the primary external baseline for policy customization, residual Q-learning). The approach is benchmarked against other variants of the algorithm on MuJoCo, with code provided, before being evaluated on the motivating use-case in Gran-Turismo Sport (GTS).

In summary, the paper has a strong problem motivation and proposed solution, but the quality and clarity of the paper could be significantly improved. As a result, this paper does not feel ready for publication yet, but I would be happy to increase my rating if these issues were improved before the end of the discussion period.

**Strengths:**

- **Motivation / Significance**
  - The problem of customizing a policy at execution time without knowledge of the original objective, and while minimising additional online data collection/fine-tuning, is an interesting and important problem.
  - This problem is well-motivated in the paper.
- **Approach / Originality**
  - The use of MPPI as a model-based approach to customize the policy at execution time seems sensible and original.
- **GTS Application**
  - The GTS results provide a clear use-case for the use of the method.
  - The videos and results appear to show a clear improvement in the policy as a result.

**Weaknesses:**

- **Quality**

  - The quality of the work should ideally be improved before the work is published. The full 10 pages are used, but there are many aspects which are unclear/undefined (see below), while there are large sections of text for which the key point is not clear. For example:
    - The discussion (first paragraph of section 4.2) should be made more concise by breaking it up into the key conclusions.
    - The abstract could also be made more concise e.g. 'Residual-MPPI can be used to improve new performance metrics at execution time, given access to the prior policy alone.'
    - The related work could similarly be made more concise by focusing on the key difference with each work.
  - Citations should be in brackets, not highlighted.
  - The appendix should come after the references in the main paper, not in the supplementary materials.
  - Other small typos and mistakes such as:
    - L416: 'Compration'
    - L361: 'apperant'

- **Clarity / Novelty of Theory**

  - The relevance and novelty of the theory is unclear to me. The approach seems to be a direct application of MPPI to Resdiaul-SAC. What is the intended contribution of Theorem 1 to the paper? It is derived in the Appendix, but the resulting Equation 5 just appears to be a direct result from [1] and it is mentioned that reference [2] already combines MPPI with SAC. Calling it a Theorem seems unnecessary if it is not a new theoretical result. The purpose of this theorem and the logic behind the derivation of Residual-MPPI could be made clearer.
  - 'Few-shot' and 'zero-shot' are mentioned throughout the paper, but not defined until Section 5.1 and it is not clear what these terms refer to until this point. These should be defined in this context before use.

- **Experiments / Significance**

  - The add-on rewards used for the MuJoCo tasks are not stated in the main paper, only the Appendix. Some examples or at least the kind of add-on tasks considered should be included in the main paper to help the reader intuitively understand what kind of customization is being considered, and to define the last but one column of Table 1, which is meaningless otherwise.

  - There are no external baselines for the MuJoCo experiements. Could residual-SAC be used as a baseline here? It is claimed that these results are zero shot, but Residual-MPPI would need a dynamics model for this to be the case. Where does the dynamics model come from for these experiments? If it is learned from data, then could Residual-SAC be trained on this data? The experimental evaluation would be strengthened by the inclusion of external baselines, or justification as to why these are not possible.

  - Residual-MPPI seems to fall between the baseline GT Sophy and Residual-SAC in terms of both Lap Time and Off-course Steps in Table 2, so Residual-SAC performs more conservatively as claimed. However, the authors mention that Residual-SAC takes much more data (80k laps) compared to Residual-MPPI (~2k laps). What would happen if Residual-SAC was trained on the equivalent smaller amount of data? I wonder if it might customize the policy less and therefore perform less conservatively on the original task, better matching the performance of Residual-MPPI? I believe this would provide a fairer comparison of the two methods so would be helpful to include if possible.

**References:**

[1] *Soft Actor-Critic: Off-Policy Maximum Entropy Deep Reinforcement Learning with a Stochastic Actor*, Haarnoja et al., 2018

[2] *Information Theoretic Model Predictive Q-Learning*, Bhardwaj et al., 2020

**Questions:**

- How were the MuJoCo add-on tasks chosen?
- How was a dynamics model obtained for Residual-MPPI for these environments?
- Why does Greedy-MPPI only degrade the performance for the Ant environment? It is stated that ignoring the log $\pi$ would degrade performance when the add-on reward is orthogonal to the basic reward in the discussion which makes sense, but why is this only the case for the Ant environment?
- Table 1 states that the evaluation results are over 500 episodes, but does not provide the number of training seeds/independent runs. How many training seeds were used? If only one seed was used, it should be stated that the std is over the evaluation period, not over different training runs, and this should be justified.
- Why are two decimal places used in Table 1? This would be clearer to the nearest integer (except the undefined last but one column).

---

> ### Author Response · Authors · 2024-11-19
> **Response to Reviewer VRbb (Part1)**
>
> Thank you for your thoughtful feedback and valuable suggestions. We are glad you found the Residual-MPPI novel and that our experiments are solid in complex environments. Please find our responses to address your concerns regarding our work below.
>
>
> ### 1: Writing and Orgnization
>
> >W1: The quality of the work should ideally be improved before the work is published. The full 10 pages are used, but there are many aspects which are unclear/undefined (see below), while there are large sections of text for which the key point is not clear.
>
> >W2.2: 'Few-shot' and 'zero-shot' are mentioned throughout the paper, but not defined until Section 5.1 and it is not clear what these terms refer to until this point. These should be defined in this context before use.
>
> > W3.1: The add-on rewards used for the MuJoCo tasks are not stated in the main paper, only the Appendix. Some examples or at least the kind of add-on tasks considered should be included in the main paper to help the reader intuitively understand what kind of customization is being considered, and to define the last but one column of Table 1, which is meaningless otherwise.
>
> >Q5: Why are two decimal places used in Table 1? This would be clearer to the nearest integer (except the undefined last but one column).
>
> Thank you for the valuable suggestions on paper organization and presentation clarity. We provide a more concise version of the discussion, abstract, and related work. The citations are now in brackets, and the appendix comes after the references in the main paper. We add a brief introduction to the exact add-on tasks before presenting the results in MuJoCo. Please find more detailed updates in the revised paper.
>
> ### 2: Novelty of Theory
>
> >W2.1: The relevance and novelty of the theory is unclear to me. The approach seems to be a direct application of MPPI to Resdiaul-SAC. What is the intended contribution of Theorem 1 to the paper? It is derived in the Appendix, but the resulting Equation 5 just appears to be a direct result from [1] and it is mentioned that reference [2] already combines MPPI with SAC. Calling it a Theorem seems unnecessary if it is not a new theoretical result. The purpose of this theorem and the logic behind the derivation of Residual-MPPI could be made clearer.
>
> Thank you for asking the questions about the Residual-MPPI theory. Please find a thorough discussion in Comment 4 in the general response.
>
> **Theorem 1 is required to validate the max-entropy property of MPPI, which is a prerequisite for the theoretical foundation established in RQL.** The augmented MDP in our formulation is derived under the assumption of max-entropy optimization. Only when optimizing a max-entropy policy on the proposed augmented MDP $\mathcal{M}^\mathrm{aug} = (\mathcal{X}, \mathcal{U}, \omega^\prime \log \pi(\boldsymbol{u}|\boldsymbol{x}) + r_R, p)$ does the resulting policy become equivalent to optimizing a max-entropy policy on the full MDP $\hat{\mathcal{M}} = (\mathcal{X}, \mathcal{U}, \omega r+r_R, p)$.
>
> **Additionally, we would like to clarify that the primary theoretical contribution of our work lies in deriving Residual-MPPI by applying max-entropy MPPI to the formulated augmented MDP.** Specifically, the reference [2] is not targeted on solving the policy customization task and can be viewed as the Valued-MPPI baseline. While the max-entropy property of MPPI has been proved in the literature [2], it is derived with a different problem formulation. Thus, we provide a self-contained and concise notation of this property in Theorem 1 and Appendix A to better serve our purpose.
>
> Since it is not our main theoretical contribution, we have renamed it to **Proposition 1** in the revised version to more accurately denote the scope of our theoretical contribution.
>
> ### 3: Additional Baselines
> >W3.2: There are no external baselines for the MuJoCo experiements. Could residual-SAC be used as a baseline here?
>
> Thank you for noting the additional learning-based baselines. We have included the results of suggested baselines in MuJoCo. Please find a complete discussion and additional results in Comment 1 in the general response and the revised Table 1 in Section 4.

---

> > ### Author Response · Authors · 2024-11-19
> > **Response to Reviewer VRbb (Part2)**
> >
> > ### 4: Data-Efficiency
> > > W3.2: If it is learned from data, then could Residual-SAC be trained on this data? The experimental evaluation would be strengthened by the inclusion of external baselines, or justification as to why these are not possible.
> >
> > > W3.3: Residual-MPPI seems to fall between the baseline GT Sophy and Residual-SAC in terms of both Lap Time and Off-course Steps in Table 2, so Residual-SAC performs more conservatively as claimed. However, the authors mention that Residual-SAC takes much more data (80k laps) compared to Residual-MPPI (~2k laps). What would happen if Residual-SAC was trained on the equivalent smaller amount of data? I wonder if it might customize the policy less and therefore perform less conservatively on the original task, better matching the performance of Residual-MPPI? I believe this would provide a fairer comparison of the two methods so would be helpful to include if possible.
> >
> > Thank you for noting the fair comparison of data-efficiency between Residual-SAC and Residual-MPPI.
> >
> > In MuJoCo, the dynamics is learned from data and Residual-SAC with the same amount of training samples used to train the dynamics model could also serve as a baseline. In the revised manuscript, we reported the performance of this baseline in Table 1 in Section 4. The results show that Residual-MPPI outperforms this baseline, demonstrating better data-efficiency of the proposed method.
> >
> > Similar results are also observed in GTS. According to our experiments in GTS, the Residual-SAC policy does not intuitively transition gradually from being aggressive to conservative during training. Instead, Residual-SAC undergoes a significant performance drop, leading to failure, and slowly recovers from it during the training process. Even though we have applied various techniques (see Appendix D.4) to enhance the training stability and final performance, we still observed a large performance drop in the initial stage of the training.  In GTS, Residual-SAC could not finish a lap at the checkpoint with 2k laps of training data (see Table 2, Section 5.2). The in-game evaluation video of the corresponding checkpoint can be found in the supplementary material.
> >
> > ### 5: Comparison between Residual-MPPI and Greedy-MPPI
> > > Q3: Why does Greedy-MPPI only degrade the performance for the Ant environment? It is stated that ignoring the $\log \pi$ would degrade performance when the add-on reward is orthogonal to the basic reward in the discussion which makes sense, but why is this only the case for the Ant environment?
> >
> > Thank you for noting the important comparison between Residual-MPPI and Greedy-MPPI. Please find a thorough comparison and discussion in Comment 3 in the general response.
> >
> > **The biggest difference is that Greedy-MPPI optimizes the objective of a biased MDP $\mathcal{M}^\mathrm{add} = (\mathcal{X}, \mathcal{U}, r_R, p)$.** As we discussed in Section 4.2, the Greedy-MPPI baseline relies solely on sampling from the prior to strike a balance between the basic and add-on tasks. In the first three tasks, where the add-on tasks are more "aligned" with the basic goal, like moving forward with less use of one joint (HalfCheetah, Swimmer) or jumping higher while jumping forward (Hopper), this biased characteristic does not lead to severe consequences and may even achieve better performance due to a simpler optimization objective and reduced reliance on biased dynamics.
> >
> > **Nevertheless, in tasks where the add-on reward is sparse or orthogonal to the basic reward, this approach would bring apparent suboptimality**. In Ant, the agent is rewarded to make progress along the x-axis in the basic task. The add-on reward further encourages the agent to make progress along the y-axis. These two reward terms are orthogonal. Solely maximizing the progress along the y-axis would sacrifice the progress along the x-axis, which explains the performance gap of Greedy-MPPI. **This issue is further exaggerated in the challenging GTS task.** In GTS, the Greedy-MPPI only optimizes the trajectories by not allowing the policy to be off-course, where a straightforward local optimal but undesirable solution would be just staying still. Such a trivial policy would behave better on the add-on tasks (stay on course) but not on the full task (stay on course while maintaining racing superiority). As shown in the Greedy-MPPI results in Figure 3 in Section 5, this bias leads to severe failure in GTS, which directly demonstrates the superiority of Residual-MPPI over Greedy-MPPI.
> >
> > The main advantage of Residual-MPPI lies in its objective, which is defined as a principled trade-off between the basic and add-on tasks. The other heuristical modifications of MPPI, including Greedy-MPPI, may work fairly well in some cases, but their performance cannot be assured. Instead, Residual-MPPI always performs equally well or even better, thanks to its theoretical ground.

---

> > > ### Author Response · Authors · 2024-11-19
> > > **Response to Reviewer VRbb (Part3)**
> > >
> > > ### 6: Details in Experiments
> > >
> > > Thank you for asking about the important details in MuJoCo and GTS experiments.
> > >
> > > > Q1: How were the MuJoCo add-on tasks chosen?
> > >
> > > In MuJoCo, we conducted experiments to validate the effectiveness and theoretical correctness of the proposed algorithm, which served as preliminary proof for further experiments in more complex tasks. We design the add-on reward to match scenarios when deploying control policies. Taking HalfCheetah as an example, we apply an extra penalty on the angle of a certain joint, which is a common issue in deployment if the corresponding motor is broken.
> > >
> > > > W3.2: It is claimed that these results are zero shot, but Residual-MPPI would need a dynamics model for this to be the case. Where does the dynamics model come from for these experiments?
> > >
> > > > Q2: How was a dynamics model obtained for Residual-MPPI for these environments?
> > >
> > > In MuJoCo, we roll out a noise-disturbed prior policy to collect training samples and train a MLP-based dynamics model by minimizing the multi-step errors on the collected data. In the Few-shot MPPI baseline in GTS, we further validated that the data from the online planning could help in fine-tuning the dynamics for better performance. In the revised manuscript, we provide the pseudocode for the complete deployment pipeline in Algorithm 2, Appendix B.
> > >
> > > To better demonstrate the importance of the selected dynamics training strategies, we have included the corresponding ablation studies in the complex GTS environment. Please find the results in Table 11 of the revised Appendix F.4.
> > >
> > > > Q4: Table 1 states that the evaluation results are over 500 episodes, but does not provide the number of training seeds/independent runs. How many training seeds were used? If only one seed was used, it should be stated that the std is over the evaluation period, not over different training runs, and this should be justified.
> > >
> > > The std is over the evaluation period since the proposed method is a planning algorithm and only one seed is used. All the baselines utilize the same prior policy and dynamics model during evaluation. Thank you for pointing it out. We have updated the corresponding experiment sections with table notes in the revised version.
> > >
> > > ### Reference
> > > [1] Soft Actor-Critic: Off-Policy Maximum Entropy Deep Reinforcement Learning with a Stochastic Actor, Haarnoja et al., 2018
> > >
> > > [2] Information Theoretic Model Predictive Q-Learning, Bhardwaj et al., 2020

---

> > > > ### Comment · Reviewer_VRbb · 2024-11-22
> > > > **Thank you to the authors for their response and improvements to the paper**
> > > >
> > > > Thank you to the authors for their thorough response. I believe that the rebuttal has significantly improved the paper. In particular, regarding the points above:
> > > >
> > > > **1. Writing and Organization:** Thank you for the substantial improvements here. The paper now appears much more ready for publication, although I believe could still be refined further to improve the clarity and make the writing more concise.
> > > >
> > > > **2. Novelty of Theory:** Thank you for the clarification and appropriately reducing the claim regarding the theoretical contribution.
> > > >
> > > > **3. Additional Baselines:** The additional Residual-SAC baselines are appreciated and make the experimental results more convincing. However, I think it could be helpful to move Residual-SAC (200K steps) above the line (and therefore included in the bolding), since this baseline is comparable, while Residual-SAC (4M steps) and Fulltask-SAC provide bounds and are not comparable and to make this clearer in the table or caption.
> > > >
> > > > **4. Data Efficiency:** This is again appreciated and makes the value of the contribution more convincing, although the presentation of Table 2 could still be improved. Personally I feel that only Residual-SAC (2K laps) is a useful addition, or that the entire second row could simply be explained in a sentence in the analysis.
> > > >
> > > > **5. Comparison between Residual-MPPI and Greedy-MPPI:** Thank you for the clarification. This is clear.
> > > >
> > > > **6. Details in Experiments:** Thank you for these clarifications which I believe also improve the paper. The variables referred to in Table 1 should ideally be included in brackets in the descriptions of the add-on tasks in the first paragraph of Section 4.1.
> > > >
> > > > As a result of the improvements, I have increased my score to reflect the fact that I believe the paper is now marginally deserving of acceptance, and thank the authors again.

---

> > > > > ### Author Response · Authors · 2024-11-26
> > > > > **Thanks for your suggestions and rasing the score**
> > > > >
> > > > > Thanks for your quick reply and rasing the score. Thanks again for your suggestion, which has been incredibly helpful in improving our work!

---

### Official Review · Reviewer_e9vs · 2024-10-24

**Soundness:** 3
**Presentation:** 4
**Contribution:** 3
**Rating:** 8
**Confidence:** 3

**Summary:**

This work approaches the challenge of policy customization: given a prior policy (trained on a task, represented by a reward function), how should one adapt it at test-time to account for a different objective (add-on reward)? To tackle this problem, the paper introduces Residual-MPPI, which bridges Residual Q-Learning (an RL algorithm for policy customization) and MPPI (a planning method for model predictive control). The work firstly introduces a theoretical motivation for this mix of techniques (Section 3.1) and then provides experiments in continuous control tasks on MuJoCo and the GTS environment. Empirical results demonstrate Residual-MPPI is capable of adapting the policy with fewer samples than prior work, and not considerably compromising performance on the prior task.

**Strengths:**

- The work is well written and structured, easy to follow. The adopted motivation and proposed method are also clear;

- The evaluation setup is interesting: it brings MuJoCo benchmarks (which, although toy problems, are not trivial for continuous control) and also on GTS (which is a more complex setup for control). Therefore, the presented results are grounded in solid benchmarks.

- The discussion in Section 4.2 is very elucidative, and convincingly justifies the failures of the baselines in comparison to the proposed method. This helps understand the strengths of Residual-MPPI.

- Noticing that it is possible to replace the terminal value estimator for the log probabilities of the policy (into the MaxEnt framework) sounds particularly clever and also justifies well the proposed method.

**Weaknesses:**

While my concerns are not major, there are some suggestions that could improve the clarity of the results in the paper:

- In Section 5.1, the paper argues that the metrics to evaluate on GTS are lap time and off-course steps since the rewards are complex. Nonetheless, I believe the work should provide both the prior task reward and the add-on reward for each method, as in the MuJoCo case, to understand how the method influences the returns that are optimized. Perhaps the considered small changes in lap time are associated with big changes in the returns, which would mean that Residual-MPPI is actually considerably deviating from the prior task designed by the environment.

- Similarly, the work could provide results for Residual SAC in the MuJoCo environments, as the most direct comparison baseline in the literature. Currently, the only results are in GTS. Providing MuJoCo results should also give a better understanding about this direct comparison.

- Another baseline that is important is training the “prior policy” on the full task (prior reward + add-on reward). Since MPPI is just a local search around the prior policy, perhaps it is too regularized towards the prior task. By providing the prior policy on the full task, the work provides a potential upper bound on joint performance, which contextualizes the performance of Residual-MPPI (as a policy customization method) to standard full RL training. This also helps clarify the trade-off of retraining a policy from scratch (expensive, but potentially better asymptotic performance) OR performing policy optimization (cheap, but potentially suboptimal).

- Lastly, the work should probably highlight more the results in terms of sample efficiency. I understand that sample efficiency is key to policy customization, particularly in the physical world. The work just mentions that MPPI leverages 2100 laps while Residual-SAC leverage 80000 laps, which is a big difference. From my understanding, this is a big contribution of the method and should be highlighted. Some suggestions are either bringing a temple of sample analysis to the main paper or at least highlighting these results in the text.


Typo: Section 5.2.1: “Compration”

**Questions:**

As mentioned in my previous points, how does Residual-MPPI performs in comparison to training a prior policy on the full task (initial task + add-on reward)?



**Summary of the Review**:

The work is well written, the method is clearly presented and well motivated, the evaluation setup is also interesting. I do believe the work could improve in some directions, particularly adding a few more baselines and highlighting some important contributions. Nonetheless, I believe it is already in a good shape for acceptance.

---

> ### Author Response · Authors · 2024-11-19
> **Response to Reviewer e9vs**
>
> Thank you for your thoughtful feedback and valuable suggestions. We are glad you found the Residual-MPPI novel and that our experiments are solid in complex environments. Please find our responses to address your concerns regarding our work below.
>
> ### 1: Reward-Level Evaluation in GTS
> >W1: In Section 5.1, the paper argues that the metrics to evaluate on GTS are lap time and off-course steps since the rewards are complex. Nonetheless, I believe the work should provide both the prior task reward and the add-on reward for each method, as in the MuJoCo case, to understand how the method influences the returns that are optimized. Perhaps the considered small changes in lap time are associated with big changes in the returns, which would mean that Residual-MPPI is actually considerably deviating from the prior task designed by the environment.
>
> Thank you for noting an additional way for evaluation in GTS. In GTS, we seek to solve a complex, realistic racing task with the proposed Residual-MPPI, where the most important metrics are lap time and off-course steps. GT Sophy 1.0 was trained to minimize the lap time. Since lap time is sparse and hard to optimize, a dense reward with multiple regularization terms was designed through reward shaping. This shaped and dense reward makes reward-level evaluation less informative and indirect. Therefore, we choose lap time and off-course steps as the evaluation metrics to quantify the performance of the policy customization in the basic and add-on tasks, respectively.
>
> Specifically, the nominal term in the basic reward is the speed along the centerline, which is correlated with the lap time. This relationship implies that a small change in the lap time indicates a minor variation in the basic reward and, consequently, a small deviation from the prior policy.
>
> ### 2: Additional Learning-based Baselines
> >W2: Similarly, the work could provide results for Residual SAC in the MuJoCo environments, as the most direct comparison baseline in the literature. Currently, the only results are in GTS. Providing MuJoCo results should also give a better understanding about this direct comparison.
>
> >W3: Another baseline that is important is training the “prior policy” on the full task (prior reward + add-on reward). Since MPPI is just a local search around the prior policy, perhaps it is too regularized towards the prior task. By providing the prior policy on the full task, the work provides a potential upper bound on joint performance, which contextualizes the performance of Residual-MPPI (as a policy customization method) to standard full RL training. This also helps clarify the trade-off of retraining a policy from scratch (expensive, but potentially better asymptotic performance) OR performing policy optimization (cheap, but potentially suboptimal).
>
> >Q1: As mentioned in my previous points, how does Residual-MPPI performs in comparison to training a prior policy on the full task (initial task + add-on reward)?
>
> Thank you for noting the additional learning-based baselines. We have included the results of the suggested baselines in MuJoCo in Table 1 in Section 4. Please find a complete discussion in Comment 1 in the general response.
>
>
> ### 3: Highlight the Sample-Efficiency
> >W4: Lastly, the work should probably highlight more the results in terms of sample efficiency. I understand that sample efficiency is key to policy customization, particularly in the physical world. The work just mentions that MPPI leverages 2100 laps while Residual-SAC leverage 80000 laps, which is a big difference. From my understanding, this is a big contribution of the method and should be highlighted. Some suggestions are either bringing a temple of sample analysis to the main paper or at least highlighting these results in the text.
>
> Thank you for noting the key advantage of our proposed methods. To highlight the sample efficiency of Residual-MPPI, we have added the results of Residual-SAC trained on the same amount of rollouts in both MuJoCo and GTS. Please find the corresponding discussion and the additional experiments in Comment 2 in the general response and revised Table 1 and Table 2 in Section 4 and Section 5.

---

> > ### Comment · Reviewer_e9vs · 2024-11-29
> > **Thank you for your response!**
> >
> > Dear authors,
> >
> > Thank you for your responses. I believe the rebuttal addressed my concerns and I also appreciate the efforts on clarifying the points raised by other reviewers. I am increasing my score to 8 since I believe the paper is in a good shape for acceptance.

---

> ### Author Response · Authors · 2024-11-23
>
> Dear Reviewer,
>
> Thank you for your feedback! We were wondering if you have gotten a chance to go over our responses  --- we have now provided clarifications and new experiment results on your concerns. We are happy to discuss further.

---

### Official Review · Reviewer_8CjN · 2024-10-27

**Soundness:** 2
**Presentation:** 3
**Contribution:** 2
**Rating:** 5
**Confidence:** 3

**Summary:**

The paper studies the problem of online-fining pre-trained policies and proposes to combine the residual q learning and Model Predictive Path Integral (MPPI) together to form a new method, called Residual-MPPI, to address the problem. The paper provides a theoretical analysis of the proposed formulation and a comprehensive empirical evaluation of the method on Mujoco and GTS domains. The results are reported to be promising and the proposed method turns out to be effective.

**Strengths:**

* [Originality] The paper attempts to integrate the residual Q learning into the Model Predictive Path Integral (MPPI) and show some promising results of such attempt.
* [Quality and clarity] The paper is well written, and the structure is good in general. It is good to see the empirical evaluation has been conducted on multiple different domains and a relatively thorough results are reported.
* [Significance] The proposed method considers an important problem of adapting a policy to new setting, which can be very useful for RL to be applied to many real-world applications.

**Weaknesses:**

### [Empirical evaluation appears to be weak]
1. The empirical performance of the proposed method, Residual-MPPI, is quite similar to the heuristic modification of MPPI, i.e., Greedy-MPPI. On the main metrics, Full-task and Add-on Task, the reported performance of Residual-MPPI is approximately the same as that of Greedy-MPPI. The paper pointed out one difference on Ant Full Task. But it is unclear how the total reward was computed here and why this total reward should matter more than the add-on reward, given that Greedy-MPPI yields higher rewards than Residual-MPPI on add-on task. So please **provide a more detailed analysis of the differences between Residual-MPPI and Greedy-MPPI, particularly in cases where their performance is similar**. Also please **clarify how the total reward is calculated and why it's considered more important than the add-on reward in certain cases, especially for the Ant Full Task**.

2. The baseline methods only cover some variants of MPPI. The paper only compared its proposed method with different variants of MPPI. However, as mentioned in the related work, there are many more baseline methods that had been discussed in contrast to the proposed method. So the paper should at least provide convincing argument that why these mentioned methods in related work should not be included as baselines. I think at least RL-driven MPPI, JSRL, and AWAC can be applied to the Mujoco environment to solve the online fine-tuning tasks. So please **include RL-driven MPPI, JSRL, and AWAC (if appropriate) to the baseline methods as they are discussed in the related work as important previous works**, or please **explain why these preivous methods in related work could not be included in the experiment**.

3. Some metrics adopted in the experiments are debatable. The paper adopted two metrics, add-on reward and basic task reward, as the main measures of the required performance. However, I’m not sure if I understand it correctly: should we always focus on the augmented reward (or the add-on reward) as the sole measure? Since this is the main objective in the policy customization. See my questions in the Question Section.

4. Results on GT Sophy and sim-to-sim experiments seem to be incomplete. It’s good to see the paper evaluates the proposed method on GT Sophy and sim-to-sim domains. But the results are relatively incomplete (e.g., no quantitative results are reported for Full, Guided, Greedy and Valued variants of MPPI). Given that the Greedy-MPPI was reported to perform similarly to the proposed method, Residual-MPPI, it would be necessary to see how Greedy performs in other domains.  So please **provide quantitative results for the missing MPPI variants in the GT Sophy and sim-to-sim experiments**.

### [More clarifications and ablations are needed to understand the method]
In section 3.2, the paper introduced three main techniques to improve the learning of the transition dynamics function. However, the paper did not explicitly describe how these techniques are used in the method. At least, it’s not apparent to find them in the algorithm 1 (can’t find the multi-step error used as a loss function). Neither is clear about fine-tuning with online data. So please **provide a more detailed explanation of how the three techniques (multi-step error, exploration noise, and fine-tuning with online data) are integrated into the algorithm, possibly by updating Algorithm 1 or providing additional pseudocode.**

Further, there are no ablation studies on many design choices reported by the paper. Specifically,  it is not reported that how the above three techniques of learning the transition dynamics can affect the empirical performance of Residual-MPPI. Also, comparing Greedy-MPPI against Residual-MPPI, it can be hard to say it is necessary to include $\log\pi$ as a reward in Residual-MPPI since the two perform very similarly on HalfCheetah, Swimmer, and Hopper.

### Clarity and the rigorousness
Some sections lacks clarity. Section 3.2: DYNAMICS LEARNING AND FINE-TUNING appear to be disjoint to Section 3.1 since the proposed techniques in Section 3.2 cannot be found in Section 3.1 or the algorithm. Re Section 5.2.2, it is vague as to how the evaluation was conducted and what quantities had been compared against to draw the conclusions there.

**Questions:**

1.  In the experiment part, why the customised the policy need to maintain the same level of performance on the basic tasks? I understand that the full task is given by $wr+r_R$ and the add-on task is given by $r_R$. It makes sense to report results of customized polices on these two tasks. But what’s the motivation of comparing the results on task defined by $r$? since it is irrelevant to the customization objective.
2. The paper formulated the policy customization as an augmented MDP, with the reward to be defined as $w’\log \pi(u|x) + r_R$. I’m wondering, in Table 1, the total reward refers to $ w’\log \pi(u|x) + r_R$ or $wr+r_R$?
3. Re Theorem 1, I can’t understand why such theorem is needed? And what does it imply?

---

> ### Author Response · Authors · 2024-11-19
> **Response to Reviewer 8CjN (Part1)**
>
> Thank you for your thoughtful feedback and valuable suggestions. We are glad you found the Residual-MPPI novel and that our experiments are solid in complex environments. Please find our responses to address your concerns regarding our work below.
>
> ### 1: Comparison between Residual-MPPI and Greedy-MPPI
> >W1.1: The empirical performance of the proposed method, Residual-MPPI, is quite similar to the heuristic modification of MPPI, i.e., Greedy-MPPI. On the main metrics, Full-task and Add-on Task, the reported performance of Residual-MPPI is approximately the same as that of Greedy-MPPI. The paper pointed out one difference on Ant Full Task. But it is unclear how the total reward was computed here and why this total reward should matter more than the add-on reward, given that Greedy-MPPI yields higher rewards than Residual-MPPI on add-on task. So please provide a more detailed analysis of the differences between Residual-MPPI and Greedy-MPPI, particularly in cases where their performance is similar.
>
> >W2.2: Also, comparing Greedy-MPPI against Residual-MPPI, it can be hard to say it is necessary to include $\log \pi$ as a reward in Residual-MPPI since the two perform very similarly on HalfCheetah, Swimmer, and Hopper.
>
> Thank you for noting the important comparison between Residual-MPPI and Greedy-MPPI. Please find a thorough comparison and discussion in Comment 3 in the general response.
>
> **The biggest difference is that Greedy-MPPI optimizes the objective of a biased MDP $\mathcal{M}^\mathrm{add} = (\mathcal{X}, \mathcal{U}, r_R, p)$.** As we discussed in Section 4.2, the Greedy-MPPI baseline relies solely on sampling from the prior to strike a balance between the basic and add-on tasks. In the first three tasks, where the add-on tasks are more "aligned" with the basic goal, like moving forward with less use of one joint (HalfCheetah, Swimmer) or jumping higher while jumping forward (Hopper), this biased characteristic does not lead to severe consequences and may even achieve better performance due to a simpler optimization objective and reduced reliance on biased dynamics.
>
> **Nevertheless, in tasks where the add-on reward is sparse or orthogonal to the basic reward, this approach would bring apparent suboptimality**. In Ant, the agent is rewarded to make progress along the x-axis in the basic task. The add-on reward further encourages the agent to make progress along the y-axis. These two reward terms are orthogonal. Solely maximizing the progress along the y-axis would sacrifice the progress along the x-axis, which explains the performance gap of Greedy-MPPI. **This issue is further exaggerated in the challenging GTS task.** In GTS, the Greedy-MPPI only optimizes the trajectories by not allowing the policy to be off-course, where a straightforward local optimal but undesirable solution would be just staying still. Such a trivial policy would behave better on the add-on tasks (stay on course) but not on the full task (stay on course while maintaining racing superiority). As shown in the Greedy-MPPI results in Figure 3 in Section 5, this bias leads to severe failure in GTS, which directly demonstrates the superiority of Residual-MPPI over Greedy-MPPI.
>
> The main advantage of Residual-MPPI lies in its objective, which is defined as a principled trade-off between the basic and add-on tasks. The other heuristical modifications of MPPI, including Greedy-MPPI, may work fairly well in some cases, but their performance cannot be assured. Instead, Residual-MPPI always performs equally well or even better, thanks to its theoretical ground.

---

> > ### Author Response · Authors · 2024-11-19
> > **Response to Reviewer 8CjN (Part2)**
> >
> > ### 2: Additional Baselines
> >
> > >W1.2: The baseline methods only cover some variants of MPPI. The paper only compared its proposed method with different variants of MPPI. However, as mentioned in the related work, there are many more baseline methods that had been discussed in contrast to the proposed method. So the paper should at least provide convincing argument that why these mentioned methods in related work should not be included as baselines. I think at least RL-driven MPPI[1], JSRL[2], and AWAC[3] can be applied to the Mujoco environment to solve the online fine-tuning tasks. So please include RL-driven MPPI, JSRL, and AWAC (if appropriate) to the baseline methods as they are discussed in the related work as important previous works, or please explain why these preivous methods in related work could not be included in the experiment.
> >
> > Thank you for noting these three related works. RL-driven MPPI [1] employs the value function of the prior policy as the terminal value estimator, which is the Valued-MPPI baseline included in the paper. In our experiments, we demonstrated the superiority of Residual-MPPI over this baseline. Regarding JSRL [2] and AWAC [3], as stated in the problem formulation, policy customization aims to **jointly optimize the unknown basic reward and the add-on reward of the full task**. In contrast, JSRL and AWAC are algorithms for RL fine-tuning, where the goal is to fine-tune a prior policy solely towards a new reward function. The prior policy is used to regularize and accelerate fine-tuning. The difference between policy customization and RL fine-tuning has been elaborated in the original RQL [4] paper. Therefore, we believe that these two works are not suitable baselines in our setting. Please find a complete discussion in Comment 1 in the general response.
> >
> > ### 3: Motivation for Policy Customization Task
> > > W1.1: Why the total reward is considered more important than the add-on reward in certain cases, especially for the Ant Full Task.
> >
> > > W1.3: Some metrics adopted in the experiments are debatable. The paper adopted two metrics, add-on reward and basic task reward, as the main measures of the required performance. However, I’m not sure if I understand it correctly: should we always focus on the augmented reward (or the add-on reward) as the sole measure? Since this is the main objective in the policy customization. See my questions in the Question Section.
> >
> > > Q1: In the experiment part, why the customised the policy need to maintain the same level of performance on the basic tasks? I understand that the full task is given by $\omega r + r_R$ and the add-on task is given by $r_R$. It makes sense to report results of customized polices on these two tasks. But what’s the motivation of comparing the results on task defined by $r$? since it is irrelevant to the customization objective.
> >
> > As we have clarified at the beginning of the general response, the goal of policy customization is to customize a given prior policy towards **jointly maximizing an add-on reward and the underlying reward of the prior policy, which is unknown in a priori**. Therefore, **the add-on reward is not the sole measure** in policy customization. The motivation is to rectify the prior policy to satisfy some additional requirements while maintaining its original characteristics and behavior.
> >
> > For example, in GTS, we would like to have a policy that can maintain the superior racing performance of GT Sophy (basic task) and stay on course (add-on task). If we only optimize towards the add-on task (i.e., preventing the policy from driving off-course), a straightforward but undesirable local optimal is staying still. This motivation behind policy customization is also common across different fields, such as preventing catastrophic forgetting in LLM during RLHF, while the basic reward information is inaccessible since the LLM is pre-trained via supervised next-token prediction.
> >
> > In the MuJoCo experiments, since we know the basic reward used to train the prior policy, we can directly evaluate the customized policies with respect to the true policy customization objective, which is to maximize the sum of the basic and add-on rewards. We reported the basic and add-on rewards achieved by the policies to reveal how the two objectives are traded off in the policies customized through different approaches.

---

> > > ### Author Response · Authors · 2024-11-19
> > > **Response to Reviewer 8CjN (Part3)**
> > >
> > > ### 4: Additional MPPI Variants in GTS
> > > >W1.4: Results on GT Sophy and sim-to-sim experiments seem to be incomplete. It’s good to see the paper evaluates the proposed method on GT Sophy and sim-to-sim domains. But the results are relatively incomplete (e.g., no quantitative results are reported for Full, Guided, Greedy and Valued variants of MPPI). Given that the Greedy-MPPI was reported to perform similarly to the proposed method, Residual-MPPI, it would be necessary to see how Greedy performs in other domains. So please provide quantitative results for the missing MPPI variants in the GT Sophy and sim-to-sim experiments.
> > >
> > > Thank you for asking this question. As we mentioned in the main paper, we only have access to the policy network of GT Sophy 1.0; it is not possible to carry out the Valued-MPPI as a baseline in GTS. Regarding Guided-MPPI and Greedy-MPPI, we have already conducted the experiments in GTS in the initial submission. The results are shown in Figure 3 in Section 5. The results show that these two baselines are unable to finish a complete lap. Moreover, despite the failed performance in MuJoCo, we further carry out Full-MPPI as a baseline in GTS. It can not complete a single lap since it has no access to the prior policy compared with Guided-MPPI, which follows our analysis.
> > >
> > > To improve the accessity of these results, we revise Table 2 in Section 5 to include those methods with \*Failed marks.
> > >
> > >
> > >
> > > ### 5: Dynamics Design
> > > > W2.1: In section 3.2, the paper introduced three main techniques to improve the learning of the transition dynamics function. However, the paper did not explicitly describe how these techniques are used in the method. At least, it’s not apparent to find them in the algorithm 1 (can’t find the multi-step error used as a loss function). Neither is clear about fine-tuning with online data. So please provide a more detailed explanation of how the three techniques (multi-step error, exploration noise, and fine-tuning with online data) are integrated into the algorithm, possibly by updating Algorithm 1 or providing additional pseudocode.
> > >
> > > > W2.2: Further, there are no ablation studies on many design choices reported by the paper. Specifically, it is not reported that how the above three techniques of learning the transition dynamics can affect the empirical performance of Residual-MPPI.
> > >
> > > In MuJoCo, we roll out a noise-disturbed prior policy to collect training samples and train a MLP-based dynamics model by minimizing the multi-step errors on the collected data. In the Few-shot MPPI baseline in GTS, we further validated that the data from the online planning could help in fine-tuning the dynamics for better performance. In the revised manuscript, we provide the pseudocode for the complete deployment pipeline in Algorithm 2, Appendix B.
> > >
> > > To better demonstrate the importance of the selected dynamics training strategies, we have included the corresponding ablation studies in the complex GTS environment. Please find the results in Table 11, Appendix F.4.
> > >
> > > ### 6: Clarity and the rigorousness
> > > > W3: Some sections lacks clarity. Section 3.2: DYNAMICS LEARNING AND FINE-TUNING appear to be disjoint to Section 3.1 since the proposed techniques in Section 3.2 cannot be found in Section 3.1 or the algorithm. Re Section 5.2.2, it is vague as to how the evaluation was conducted and what quantities had been compared against to draw the conclusions there.
> > >
> > > Thank you for your suggestions on paper clarity. We have updated the corresponding sections to improve the clarity and consistency in the revised version.
> > >
> > > ### 7: Details in MuJoCo Experiments
> > > > Q2: The paper formulated the policy customization as an augmented MDP, with the reward to be defined as $\omega^\prime \log \pi(u | x) + r_R$. I’m wondering, in Table 1, the total reward refers to $\omega^\prime \log \pi(u | x) + r_R$ or $\omega r + r_R$?
> > >
> > > In Table 1, the total reward refers to $\omega r + r_R$ since it is the target task we want to optimize. Thank you for pointing this out; we have revised Table 1 in Section 4 to include this detail.

---

> > > > ### Author Response · Authors · 2024-11-19
> > > > **Response to Reviewer 8CjN (Part4)**
> > > >
> > > > ### 8: Questions on Theorem 1
> > > > > Q3: Re Theorem 1, I can’t understand why such theorem is needed? And what does it imply?
> > > >
> > > > Thank you for asking the questions about the Residual-MPPI theory. Please find a thorough discussion in Comment 4 in the general response.
> > > >
> > > > **Theorem 1 is required to validate the max-entropy property of MPPI, which is a prerequisite for the theoretical foundation established in RQL.** The augmented MDP in our formulation is derived under the assumption of max-entropy optimization. Only when optimizing a max-entropy policy on the proposed augmented MDP $\mathcal{M}^\mathrm{aug} = (\mathcal{X}, \mathcal{U}, \omega^\prime \log \pi(\boldsymbol{u}|\boldsymbol{x}) + r_R, p)$ does the resulting policy become equivalent to optimizing a max-entropy policy on the full MDP $\hat{\mathcal{M}} = (\mathcal{X}, \mathcal{U}, \omega r+r_R, p)$.
> > > >
> > > > Since it is not our main theoretical contribution, we have renamed it to **Proposition 1** in the revised version to more accurately denote the scope of our theoretical contribution.
> > > >
> > > > ### Reference
> > > >
> > > > [1] Qu Y, Chu H, Gao S, et al. Rl-driven mppi: Accelerating online control laws calculation with offline policy[J]. IEEE Transactions on Intelligent Vehicles, 2023.
> > > >
> > > > [2] Uchendu I, Xiao T, Lu Y, et al. Jump-start reinforcement learning[C]//International Conference on Machine Learning. PMLR, 2023: 34556-34583.
> > > >
> > > > [3] Nair A, Gupta A, Dalal M, et al. Awac: Accelerating online reinforcement learning with offline datasets[J]. arXiv preprint arXiv:2006.09359, 2020.
> > > >
> > > > [4] Li C, Tang C, Nishimura H, et al. Residual q-learning: Offline and online policy customization without value[J]. Advances in Neural Information Processing Systems, 2023, 36: 61857-61869.

---

> > > > ### Comment · Reviewer_8CjN · 2024-11-20
> > > > **Thanks for the response**
> > > >
> > > > Re "we only have access to the policy network of GT Sophy 1.0; it is not possible to carry out the Valued-MPPI as a baseline in GTS", so given a policy network, it's not possible to collect samples and then train a value network in GT Sophy 1.0?

---

> > > > > ### Author Response · Authors · 2024-11-21
> > > > > **Response to Reviewer 8CjN**
> > > > >
> > > > > Thank you for your quick reply and ackonlegament of our response.
> > > > >
> > > > > > Q1: I understand that the goal of policy customization is to jointly maximizing an add-on reward and the underlying reward of the prior policy. But my question was why the customised the policy need to maintain the same level of performance on the basic tasks?
> > > > >
> > > > > The customized policy indeed does not necessarily maintain the same level of performance on the basic task. The main evaluation metric for policy customization should be the total reward, i.e., $\omega r + r_R$. When the add-on task contradicts the basic task, the optimal policy may trade off its performance on the basic task to maximize the total reward.
> > > > >
> > > > > The reason we want to measure the basic and add-on rewards separately is to monitor the trade-off achieved by the customized policy, i.e., how well the performance on the basic task is maintained and how much the policy is optimized toward the add-on task.
> > > > >
> > > > > We have revised the **Metric** paragraph of the 4.1 Experiment Setup Section in the paper to reflect this point. Thank you for pointing it out and helping us improve the clarity of our paper!
> > > > >
> > > > > > Q2:  If "the goal (of JSRL and AWAC) is to fine-tune a prior policy solely towards a new reward function", can we then make a simple modification to these two methods by fine-tuning a prior poilcy towards a joint reward function? i.e., wr + r_R
> > > > >
> > > > > In policy customization, we consider cases where we are only given a prior policy without the basic reward information. Therefore, it is infeasible to fine-tune a prior policy towards $\omega r + r_R$ in our setting. We may consider a policy fine-tuned on $\omega r + r_R$ with JSRL, AWAR, or other RL fine-tuning methods as setting a reference value, when priviledged access to the basic reward is granted. However, we believe that the Fulltask SAC result, which we added during rebuttal in response to other reviewers' requests, already serves the purpose well.
> > > > >
> > > > > > Q3:  Re "we only have access to the policy network of GT Sophy 1.0; it is not possible to carry out the Valued-MPPI as a baseline in GTS", so given a policy network, it's not possible to collect samples and then train a value network in GT Sophy 1.0?
> > > > >
> > > > > Since we consider cases without the basic reward information during policy customization, we are not able to collect samples with GT Sophy 1.0 and train a value network. Therefore, we cannot construct Valued-MPPI as a fair and feasible baseline for policy customization. Nevertheless, in MoJoCo, we still added Valued-MPPI as a reference with priviledged access to the value function of the prior policy, since it represents a typical scheme to combine a model-free policy with model-based planning (e.g., MuZero, TD-MPC, MBOP, RL-driven MPPI). In GTS, since we don't have access to the value network of GT Sophy 1.0, it is not possible to keep the same practice in GTS. While we may train a value network in a post-hoc manner by assuming priviledge access to the basic reward function, the  value network trained in this way is only trained on offline data collected by the final GT Sophy 1.0, instead of being iteratively updated using the replay buffer updated throughout policy training. The Valued-MPPI baseline constructed in this way will not be the exact form it is designed for. Therefore, we end up not including Valued-MPPI as a baseline in GTS.

---

> > > > > ### Author Response · Authors · 2024-11-26
> > > > >
> > > > > Thanks again for your prompt feedback! We were wondering if you have gotten a chance to go over our follow-up responses --- we have now provided clarifications on your concerns. We are happy to discuss further.

---

> > > ### Comment · Reviewer_8CjN · 2024-11-20
> > > **Thanks for the response**
> > >
> > > Thank you very much for the very detailed response. If "the goal (of JSRL and AWAC) is to fine-tune a prior policy solely towards a new reward function", can we then make a simple modification to these two methods by fine-tuning a prior poilcy towards a joint reward function? i.e., $wr + r_R$

---

> ### Comment · Reviewer_8CjN · 2024-11-20
> **Thanks for the response**
>
> I understand that the goal of policy customization is to **jointly maximizing an add-on reward and the underlying reward of the prior policy**. But my question was why **the customised the policy need to maintain the same level of performance on the basic tasks**?

---

### Official Review · Reviewer_c2uW · 2024-10-29

**Soundness:** 3
**Presentation:** 3
**Contribution:** 3
**Rating:** 6
**Confidence:** 4

**Summary:**

The paper introduces Residual-MPPI, a method for online policy customization in continuous control tasks. Residual-MPPI enables immediate policy adaptation at execution time by utilizing only the action distribution of a pre-trained policy, without needing additional training data or detailed knowledge of the original policy. Experiments demonstrate its effectiveness in adapting the champion-level racing agent GT Sophy 1.0 to custom constraints within the challenging Gran Turismo Sport (GTS) simulator environment, as well as in the robotic MuJoCo environment.

**Strengths:**

- The paper is well-written and easy to follow.
- The methodology is novel and well-structured.
- The approach is validated in complex environments, including MuJoCo and GTS.

**Weaknesses:**

- The paper lacks SOTA baselines for policy customization. For instance, methods like those in [1, 2] demonstrate few-shot adaptability to new environments without an additional parameter training phase. A comparison with such method would strengthen the evaluation.

[1] Xu, Mengdi, et al. "Prompting decision transformer for few-shot policy generalization." *international conference on machine learning*. PMLR, 2022.

[2] Liu, Jinxin, et al. "Ceil: Generalized contextual imitation learning." *Advances in Neural Information Processing Systems* 36 (2023): 75491-75516.

- The performance difference between Residual-MPPI and the comparison baseline, Greedy-MPPI, is not sufficiently analyzed. In Table 1, Residual-MPPI and Greedy-MPPI show very similar performance, yet the paper only briefly mentions on Line 430 that “Greedy-MPPI, as mentioned above, also leads to severe failure in complex GTS tasks” without further explanation. A more detailed analysis of Off-course Steps across different baselines would add clarity to the comparison.

**Questions:**

- I am curious about the performance difference caused by the inclusion of $\log \pi$. How does the trajectory change when the hyperparameter $\omega'$ is increased or decreased?
- When performing policy customization without a given reward, how can the proposed Residual-MPPI be applied if only expert data or instructions are provided instead?
- I am willing to raise the score if the authors respond to the weaknesses and questions.

---

> ### Author Response · Authors · 2024-11-19
> **Response to Reviewer c2uW (Part1)**
>
> Thank you for your thoughtful feedback and valuable suggestions. We are glad you found the Residual-MPPI novel and that our experiments are solid in complex environments. Please find our responses to address your concerns regarding our work below.
>
> ### 1: Additional Baselines
> > W1: The paper lacks SOTA baselines for policy customization. For instance, methods like those in [1, 2] demonstrate few-shot adaptability to new environments without an additional parameter training phase. A comparison with such method would strengthen the evaluation.
>
> Thank you for noting these two related works. In our experiments, we have conducted comparisons under the planning scope with various SOTA approaches with different level of assmuptions. Please find a complete discussion in Comment 1 in the general response.
>
> **Regarding the two mentioned works, despite their few-shot adaptability to new environments, we believe they are not appropriate baselines for the policy customization task.** Prompt-DT  [1] designs a Decision-Transformer conditioned on sequences of states, actions, and rewards to capture information about the new task, achieving "few-shot" adaptability. Ceil [2] introduces an additional contextual variable $z$ and the corresponding contextual policy $\pi(a|s, z)$ to recover the expert data information via the learned $z^*$. Their goal is to construct a policy that can adapt to a new task with few demonstrations. In contrast, policy customization aims to **customize a given prior policy towards jointly maximizing an add-on reward and the underlying reward of the prior policy, which is unknown in a priori**. This task is not solely about adapting the prior policy to a new task but also ensuring that the customized policy inherits the characteristics of the prior policy. Moreover, both Prompt-DT and Ceil rely on expert demonstrations of the new task to enable adaptation, which is unavailable in our online policy customization setting. We have added these two works and the discussion above in the revised Section 6.
>
> ### 2: Comparison between Residual-MPPI and Greedy-MPPI
> > W2: The performance difference between Residual-MPPI and the comparison baseline, Greedy-MPPI, is not sufficiently analyzed. In Table 1, Residual-MPPI and Greedy-MPPI show very similar performance, yet the paper only briefly mentions on Line 430 that “Greedy-MPPI, as mentioned above, also leads to severe failure in complex GTS tasks” without further explanation. A more detailed analysis of Off-course Steps across different baselines would add clarity to the comparison.
>
> > Q1: I am curious about the performance difference caused by the inclusion of $\log \pi$. How does the trajectory change when the hyperparameter is increased or decreased?
>
> Thank you for noting the important comparison between Residual-MPPI and Greedy-MPPI. The major difference between the Residual-MPPI and Greedy-MPPI is the inclusion of $\log \pi$, where the Greedy-MPPI can be seen as Residual-MPPI with $\omega^\prime = 0$.
>
> Theoretically, $\omega^\prime$ represents the trade-off between the basic and add-on rewards. In GTS, larger $\omega^\prime$ should lead to a behavior closer to the prior policy with shorter lap time and more off-course steps. Conversely, an excessively small $\omega^\prime$ leads to a reward term that is overly biased towards the off-course penalty. As a result, the customized policy tends to deviate too much from the optimal racing line taken by GT Sophy, leading to suboptimal performance in both lap time and off-course steps. We have conducted the corresponding ablation studies in GTS, which follow our analysis here. Please find a thorough comparison and discussion in Comment 3 of the general response and the ablation results in Appendix F.2 and F.3 in the revised paper.
>
>
> ### 3: Approach for Expert-Data-Only Customization
> > Q2: When performing policy customization without a given reward, how can the proposed Residual-MPPI be applied if only expert data or instructions are provided instead?
>
> Thank you for noting this question. In this work, we focus on applications where an add-on reward function is defined and study how to solve the policy customization problem in a sampling-efficient manner online for continuous control problems. It would be an interesting future direction to further extend Residual-MPPI to cases where only expert data or instructions are provided. If expert demonstration samples are provided, inverse RL can be adopted to infer the add-on reward from the demonstration. For instance, MEReQ [3] has explored the synergy of inverse RL and RQL. If instructions are provided, the add-on reward model can be learned from human or AI feedback [4]. We plan to explore these extensions of Residual-MPPI in our future work.

---

> ### Author Response · Authors · 2024-11-19
> **Response to Reviewer c2uW (Part2)**
>
> ### Reference
>
> [1] Xu, Mengdi, et al. "Prompting decision transformer for few-shot policy generalization." international conference on machine learning. PMLR, 2022.
>
> [2] Liu, Jinxin, et al. "Ceil: Generalized contextual imitation learning." Advances in Neural Information Processing Systems 36 (2023): 75491-75516.
>
> [3] Chen Y, Tang C, Li C, et al. MEReQ: Max-Ent Residual-Q Inverse RL for Sample-Efficient Alignment from Intervention[J]. arXiv preprint arXiv:2406.16258, 2024.
>
> [4] Ziegler D M, Stiennon N, Wu J, et al. Fine-tuning language models from human preferences[J]. arXiv preprint arXiv:1909.08593, 2019.

---

> ### Comment · Reviewer_c2uW · 2024-11-20
> **Response to author**
>
> Thank you for your thoughtful response.
>
> **Additional Baselines**
> I think that adding additional external baselines in the experiments is necessary to justify the performance improvements of the Residual-MPPI.
> Regarding Prompt-DT [1], I agree that additional datasets are required. However, as demonstrated in Figure 3 of [1], the dataset used for PromptDT does not necessarily need to be of expert-level quality.
> In this study, the experiments involve a setup where add-on tasks are appended to a basic task. Therefore, even using few-shot data collected with a policy trained on the basic task, the dataset quality is expected to be comparable to the medium dataset used in [1]. Thus, it seems feasible to apply Prompt-DT in this context.
>
> [1] Xu, Mengdi, et al. "Prompting decision transformer for few-shot policy generalization." international conference on machine learning. PMLR, 2022.

---

> > ### Author Response · Authors · 2024-11-23
> > **Response to Reviewer c2uW (Part1)**
> >
> > > Q1: I think that adding additional external baselines in the experiments is necessary to justify the performance improvements of the Residual-MPPI. Regarding Prompt-DT [1], I agree that additional datasets are required. However, as demonstrated in Figure 3 of [1], the dataset used for PromptDT does not necessarily need to be of expert-level quality.
> > In this study, the experiments involve a setup where add-on tasks are appended to a basic task. Therefore, even using few-shot data collected with a policy trained on the basic task, the dataset quality is expected to be comparable to the medium dataset used in [1]. Thus, it seems feasible to apply Prompt-DT in this context.
> >
> > Thank you for your quick reply and acknowledgement of our response. We would like to further discuss the difference between the policy customization problem solved by Residual-MPPI and the few-shot adaptation problem solved by Prompt-DT [1].
> >
> >
> > Given a prior policy, policy customization seeks to find a policy that **jointly maximizes a new add-on reward and the basic reward that the prior policy is trained to maximize**. Conversely, few-shot adaptation seeks to adapt a policy to the new reward.
> >
> > Furthermore, Residual-MPPI and Prompt-DT have very different assumptions on the prior policy being adapted. Residual-MPPI assumes the prior policy **following the maximum-entropy principle**, without specifying a particular model architecture or training algorithm. Conversely, the few-shot adaptation scheme proposed in Prompt-DT relies on a **DecisionTransformer-based prior policy trained on multi-task expert demonstration data via offline RL**. Specifically, the amount of expert-level data used in Prompt-DT training is 45 times that used in Residual-MPPI dynamics learning.
> >
> > In addition, the trajectory prompts used in Prompt-DT require **full knowledge of the reward function of the target task to adapt the policy to**. When Prompt-DT is applied to solve policy customization (i.e., adapting the policy to jointly maximize the add-on and basic rewards), we are not able **to generate accurate trajectory prompts**, since don't have access to the basic reward in policy customization.

---

> > > ### Author Response · Authors · 2024-11-23
> > > **Response to Reviewer c2uW (Part2)**
> > >
> > > Despite all these distinctions in problem settings and assumptions, we have attempted to bring Prompt-DT closer to the policy customization setting to make the comparison as fair as possible. We select the ANT-dir environment and test Prompt-DT with the official implementation and its training parameters. Similar to the Ant environment in our original experiments, we define the basic reward as the progress along the x-axis, and the add-on reward as the progress along the y-axis.
> > >
> > > First, we train a Prompt-DT model through multi-task offline RL as the prior policy. The multi-task demonstration data consists of expert trajectories walking along various directions (see *Prompt-DT Basic Task Demo* in Table 1 for the rewards of the demo trajectories provided for the basic task). Prompt-DT can be directed to solve the basic task, by conditioning on a trajectory prompt that is 1) sampled from the basic task demo, and 2) labeled with the basic reward function (see *Prompt-DT on Basic Task* in Table 1 for its performance).
> > >
> > > We construct two types of prompts to adapt Prompt-DT to the full task defined by the sum of basic and add-on rewards:
> > >
> > > - **Expert Prompt**: the trajectory prompt that is 1) collected with an *expert policy* trained on the full task, and 2) labeled with the *total reward* function. This prompt follows the original Prompt-DT setting. However, it has priviledged access to an expert policy and the total reward function, which are not given in policy customization.
> > > - **Prior Prompt**: the trajectory prompt that is 1) sampled from the basic task demo, and 2) labeled with the add-on reward function. It is the modification suggested by the reviewer to bring Prompt-DT closer to our setting.
> > >
> > > The performance of the customized policies is reported in Table 2. If provided with the prior prompt, Prompt-DT can hardly customize itself to the full task, with minor improvements on the add-on task. Moreover, even when provided with the expert prompt, Prompt-DT fails to strike a good trade-off----it sacrifices too much on the basic reward to get a higher add-on reward. The total reward is even lower than the case with the prior prompt. In contrast, Residua-MPPI customizes the prior policy to achieve a higher add-on reward with minor degradation on the basic task, leading to a much higher total reward.
> > >
> > >
> > > Prior Policy                             | Total Reward       | Basic Reward       | Add-on Reward      |
> > > --------------------------------         | ------------------ | -----------------  | -----------------  |
> > > Residual-MPPI  Prior                     | 784.9 $\pm$ 127.4  | 799.7 $\pm$ 108.5  | -14.8 $\pm$ 58.5   |
> > > Prompt-DT Basic Task Demo                | 779.5 $\pm$ 109.0  | 793.0 $\pm$ 61.7   |  -13.5 $\pm$ 25.0  |
> > > Prompt-DT on Basic Task                  | 686.4 $\pm$ 109.0  | 691.8 $\pm$ 101.1  |  -5.4 $\pm$ 45.5   |
> > >
> > >
> > >
> > > Customized Policy                        | Total Reward       | Basic Reward       | Add-on Reward      |
> > > --------------------------------         | ------------------ | -----------------  | -----------------  |
> > > Prompt-DT (Prior Prompt)                 | 678.4 $\pm$ 126.5  | 677.0 $\pm$ 114.3  |  1.3 $\pm$ 46.3    |
> > > Prompt-DT (Expert Prompt)                | 626.0 $\pm$ 187.8  | 605.6 $\pm$ 184.8  |  20.4 $\pm$ 55.5   |
> > > Residual-MPPI                            | 872.0 $\pm$ 69.0   | 774.8 $\pm$ 41.8   | 97.1 $\pm$ 48.5    |
> > >
> > > While Prompt-DT is not a fair baseline given the difference in problem settings and basic assumptions, the aforementioned discussion and comparison between Residual-MPPI and Prompt-DT could be interesting to the readers. We will include these results and discussion in the appendix at the end of the rebuttal period.
> > >
> > > ### Reference
> > > [1] Xu, Mengdi, et al. "Prompting decision transformer for few-shot policy generalization." international conference on machine learning. PMLR, 2022.

---

> ### Comment · Reviewer_c2uW · 2024-11-24
> **Thank you for response**
>
> Thank you for the detailed response. I have adjusted the score based on the author's feedback.

---

> > ### Author Response · Authors · 2024-11-26
> > **Thanks for your suggestions and rasing the score**
> >
> > Thanks for your quick reply and rasing the score. Your suggestions has been incredibly helpful in improving our work!

---

### Author Response · Authors · 2024-11-19
**Author General Response to Common Concerns (Part1)**

We thank the reviewers for their thoughtful feedback and suggestions. We are delighted that all the reviewers found the Residual-MPPI framework novel and the majority of the reviewers found it essential (R2, R3, R4). We also appreciate the reviewers' acknowledgment of the thorough empirical evaluation (R1, R2, R3, R4) and clarity of the presentation (R1, R2, R3). In this thread, we summarize our responses to the common concerns shared by the reviewers. For a reviewer-specific response, please refer to the rebuttal attached to each reviewer's comments.

### Clarification of Problem Formulation
Firstly, we want to emphasize that Residual-MPPI considers a novel problem setting that has not been addressed in the literature before. Specifically, Residual-MPPI aims to solve the policy customization task, where the goal is to **customize a given prior policy towards jointly maximizing an add-on reward and the underlying reward of the prior policy, which is unknown in a priori**. The policy customization problem was first formulated in RQL [12], where the authors proposed initial solutions leveraging model-free RL, as well as a model-based online solution based on MCTS for discrete action spaces. In Residual-MPPI, we aim to further solve the policy customization problem in a sampling-efficient manner online for continuous control problems.

### 1: Baselines Comparison

Some reviewers suggest some existing approaches in the literature that could potentially solve the policy customization task and are not compared against Residual-MPPI in our work. We would like to thank the reviewers for pointing us to these relevant papers in the literature, including those baselines already covered in the paper and those suggested by the reviewers. They are worth discussing and comparing in our context. In this section, we provide a thorough comparison between Residual-MPPI and these relevant works in the literature.


**First, the MPPI variants we covered in experiments can represent the performance of various planning approaches**. **Full-MPPI** represents the performance of the commonly used sampling-based MPC approach, MPPI [1], to maximize the total reward without a prior policy. **Guided-MPPI** further incorporates the prior policy during the sampling stage, which is similar to one ablation version of RL-driven MPPI [8]. **Valued-MPPI** assumes access to the value function together with the prior policy, representing the performance of the most commonly used SOTA approaches, such as MuZero [2], TD-MPC [3, 4], MBOP [5], and RL-driven MPPI [8]. **Greedy-MPPI**, in contrast to the previous variants, requires no knowledge of the basic reward and only maximizes the add-on reward without utilizing the $\log \pi$ term, directly serving as a baseline to evaluate the importance of the $\log \pi$ term in Residual-MPPI.

**Second, some reviewers suggested RL fine-tuning/ Few-shot adaptation works [6, 7, 9, 10] as baselines. However, we want to emphasize that their problem settings are different; thus, they are not appropriate baselines in our setting**. As stated in the problem formulation, policy customization aims to jointly optimize the unknown basic reward and the add-on reward of the full task. In contrast, RL fine-tuning algorithms, such as JSRL [9] and AWAC [10], aim to fine-tune a prior policy solely towards a new reward function. The prior policy is used to regularize and accelerate fine-tuning. The difference between policy customization and RL fine-tuning has been elaborated in the original RQL paper. Similarly, the goal of the suggested few-shot adaptation works, Ceil [6] and Prompt-DT [7] are also to adapt the prior policy solely toward a new task. Moreover, Ceil [6] and Prompt-DT [7] require an extra demonstration dataset to adapt the policy to the new task, while such expert demonstrations are unavailable in the online policy customization setting. Therefore, we believe the mentioned works are not appropriate baselines to include.


**Finally, some reviewers have suggested adding Residual-SAC and RL policies trained on Fulltask as baselines in the experiments.** In response to the requests from R3 and R4, we added Residual-SAC as a baseline in MuJoCo. Furthermore, in both MoJoCo and GTS experiments, we reported the performance of Residual-SAC trained with the same amount of data used for dynamics model training in Residual-MPPI to illustrate the superior sample efficiency of Residual-MPPI. As suggested by R3, the RL policy trained on Fulltask can serve as a performance upper bound of the policy customization task. We have included this baseline in the MoJoCo experiment (see Table 1, Section 4).

---

> ### Author Response · Authors · 2024-11-19
> **Author General Rebuttal to Common Concerns (Part2)**
>
> ### 2: Data-Efficiency
> Some reviewers (R3, R4) mentioned the fair comparison by using the same amount of data in the learning-based baselines, especially in GTS. It is more solid to demonstrate the data-efficiency of the proposed method by showing what would happen if Residual-SAC was trained on the equivalent smaller amount of data, which is from ~80k laps to ~2k laps.
>
> According to our experiments in GTS, the Residual-SAC policy does not intuitively transition gradually from being aggressive to conservative during training. Instead, Residual-SAC undergoes a significant performance drop, leading to failure, and slowly recovers from it during the training process. Even though we have applied various techniques (see Appendix D.4) to enhance the training stability and final performance, we still observed a large performance drop in the initial stage of the training.
>
> In the revised manuscript, to highlight the outstanding data-efficiency of Residual-MPPI, we reported the performance of the Residual-SAC with the same amount of training samples used to train the dynamics model in Residual-MPPI. In GTS, Residual-SAC could not finish a lap at the checkpoint with 2k laps of training data (see Table 2, Section 5.2). The in-game evaluation video of the corresponding checkpoint can be found in the supplementary material. The MuJoCo results can be found in Table 1, Section 4.
>
>
> ### 3: Comparison between Greedy-MPPI and Residual-MPPI
> Some reviewers (R1, R2, R4) raise their questions on the performance and analysis between the Greedy-MPPI and Residual-MPPI.
>
> **The biggest difference is that Greedy-MPPI optimizes the objective of a biased MDP $\mathcal{M}^\mathrm{add} = (\mathcal{X}, \mathcal{U}, r_R, p)$.** As we discussed in Section 4.2, the Greedy-MPPI baseline relies solely on sampling from the prior to strike a balance between the basic and add-on tasks. In the first three tasks, where the add-on tasks are more "aligned" with the basic goal, like moving forward with less use of one joint (HalfCheetah, Swimmer) or jumping higher while jumping forward (Hopper), this biased characteristic does not lead to severe consequences and may even achieve better performance due to a simpler optimization objective and reduced reliance on biased dynamics.
>
> **Nevertheless, in tasks where the add-on reward is sparse or orthogonal to the basic reward, this approach would bring apparent suboptimality**. In Ant, the agent is rewarded to make progress along the x-axis in the basic task. The add-on reward further encourages the agent to make progress along the y-axis. These two reward terms are orthogonal. Solely maximizing the progress along the y-axis would sacrifice the progress along the x-axis, which explains the performance gap of Greedy-MPPI. **This issue is further exaggerated in the challenging GTS task.** In GTS, the Greedy-MPPI only optimizes the trajectories by not allowing the policy to be off-course, where a straightforward local optimal but undesirable solution would be just staying still. Such a trivial policy would behave better on the add-on tasks (stay on course) but not on the full task (stay on course while maintaining racing superiority). As shown in the Greedy-MPPI results in Figure 3 in Section 5, this bias leads to severe failure in GTS, which directly demonstrates the superiority of Residual-MPPI over Greedy-MPPI.
>
> The main advantage of Residual-MPPI lies in its objective, which is defined as a principled trade-off between the basic and add-on tasks. The other heuristical modifications of MPPI, including Greedy-MPPI, may work fairly well in some cases, but their performance cannot be assured. Instead, Residual-MPPI always performs equally well or even better, thanks to its theoretical ground.
>
> Following R1's suggestions, we include an ablation study on $\omega^\prime$ to more clearly show the comparison against Greedy-MPPI. Greedy-MPPI can be seen as Residual-MPPI with $\omega^\prime = 0$, and $\omega^\prime$ represents the trade-off between the basic task and the add-on task, where larger $\omega^\prime$ should lead to a behavior closer to the prior policy. We have conducted the corresponding ablation studies in MuJoCo and GTS, which follow our analysis. Please find a more detailed discussion and the ablation results in Appendix F.2 and F.3 in the revised paper.

---

> > ### Author Response · Authors · 2024-11-19
> > **Author General Rebuttal to Common Concerns (Part3)**
> >
> > ### 4: The Necessity of Theorem 1
> > Some reviewers (R2, R4) raised questions about the necessity of Theorem 1.
> >
> > **Theorem 1 is required to validate the max-entropy property of MPPI, which is a prerequisite for the theoretical foundation established in RQL.** The augmented MDP in our formulation is derived under the assumption of max-entropy optimization. Only when optimizing a max-entropy policy on the proposed augmented MDP $\mathcal{M}^\mathrm{aug} = (\mathcal{X}, \mathcal{U}, \omega^\prime \log \pi(\boldsymbol{u}|\boldsymbol{x}) + r_R, p)$ does the resulting policy become equivalent to optimizing a max-entropy policy on the full MDP $\hat{\mathcal{M}} = (\mathcal{X}, \mathcal{U}, \omega r+r_R, p)$.
> >
> > **Additionally, we would like to clarify that the primary theoretical contribution of our work lies in deriving Residual-MPPI by applying max-entropy MPPI to the formulated augmented MDP.** While the max-entropy property of MPPI has been proved in the literature [11], it is derived with a different problem formulation. Thus, we provide a self-contained and concise notation of this property in Theorem 1 and Appendix A to better serve our purpose. Since it is not our main theoretical contribution, we have renamed it to **Proposition 1** in the revised version to more accurately denote the scope of our theoretical contribution.
> >
> > ### Reference
> > [1] Williams G, Wagener N, Goldfain B, et al. Information theoretic mpc for model-based reinforcement learning[C]//2017 IEEE international conference on robotics and automation (ICRA). IEEE, 2017: 1714-1721.
> >
> > [2] Schrittwieser J, Antonoglou I, Hubert T, et al. Mastering atari, go, chess and shogi by planning with a learned model[J]. Nature, 2020, 588(7839): 604-609.
> >
> > [3] Hansen N, Wang X, Su H. Temporal difference learning for model predictive control[J]. arXiv preprint arXiv:2203.04955, 2022.
> >
> > [4] Hansen N, Su H, Wang X. Td-mpc2: Scalable, robust world models for continuous control[J]. arXiv preprint arXiv:2310.16828, 2023.
> >
> > [5] Argenson A, Dulac-Arnold G. Model-based offline planning[J]. arXiv preprint arXiv:2008.05556, 2020.
> >
> > [6] Xu, Mengdi, et al. "Prompting decision transformer for few-shot policy generalization." international conference on machine learning. PMLR, 2022.
> >
> > [7] Liu, Jinxin, et al. "Ceil: Generalized contextual imitation learning." Advances in Neural Information Processing Systems 36 (2023): 75491-75516.
> >
> > [8] Qu Y, Chu H, Gao S, et al. Rl-driven mppi: Accelerating online control laws calculation with offline policy[J]. IEEE Transactions on Intelligent Vehicles, 2023.
> >
> > [9] Uchendu I, Xiao T, Lu Y, et al. Jump-start reinforcement learning[C]//International Conference on Machine Learning. PMLR, 2023: 34556-34583.
> >
> > [10] Nair A, Gupta A, Dalal M, et al. Awac: Accelerating online reinforcement learning with offline datasets[J]. arXiv preprint arXiv:2006.09359, 2020.
> >
> > [11] Bhardwaj M, Handa A, Fox D, et al. Information theoretic model predictive q-learning[C]//Learning for Dynamics and Control. PMLR, 2020: 840-850.
> >
> > [12] Li C, Tang C, Nishimura H, et al. Residual q-learning: Offline and online policy customization without value[J]. Advances in Neural Information Processing Systems, 2023, 36: 61857-61869.

---

### Meta-Review · Area_Chair_dyF6 · 2024-12-21

**Metareview:**

The paper proposes Residual-MPPI, a method for online policy customization in continuous control tasks. The proposed method bridges model predictive path integral (MPPI) control and residual Q-Learning to adapt a pre-trained policy to new objectives at execution time without additional training on the prior task. The experiments conducted in the MuJoCo and Gran Turismo Sport (GTS) environments demonstrate sample-efficient adaptation.

Reasons to accept
- Generally well-written and structured, with a clear problem statement and motivation.
- This work tackles the challenging problem of policy customization at execution time without fine-tuning or detailed knowledge of the original reward function.
- The proposed method combines MPPI and residual Q-Learning in a novel way for policy customization.
- The evaluation is comprehensive, covering diverse benchmarks (MuJoCo and GTS).
- The experimental results demonstrate significantly improved sample efficiency of the proposed method compared to baselines.

Reasons to reject
- Missing ablation studies on many design choices. (mostly addressed by the rebuttal)
- Residual-MPPI performs similarly to Greedy-MPPI in many tasks, but there is insufficient analysis of these results. (addressed by the rebuttal)
- Missing results for a "prior policy" trained on the full task, which could serve as an upper performance bound. (addressed by the rebuttal)
- Terms like "few-shot" and "zero-shot" are introduced late without adequate explanation.
- Details on how dynamics models are trained and used are sparse. (mostly addressed by the rebuttal)

Overall, this paper studies a promising research direction and presents an interesting and effective approach. Many of the concerns and questions raised by the reviewers were sufficiently addressed by the author's rebuttal and in the revised paper. Consequently, I recommend accepting the paper.

**Additional Comments On Reviewer Discussion:**

During the rebuttal period, all four reviewers acknowledged the author's rebuttal, and three reviewers adjusted the score accordingly.

---

### Decision · Program_Chairs · 2025-01-22

Accept (Poster)